# A linear assessment of barotropic Rossby wave propagation in different background flow configurations

Antonio Segalini[1], Jacopo Riboldi[1], Volkmar Wirth[2], and Gabriele Messori[1,3,4,5]

[1]Department of Earth Sciences, Uppsala University, Villavägen 16, 752 36 Uppsala, Sweden
[2]Institute for Atmospheric Physics, Johannes Gutenberg University Mainz, Becherweg 21, 55128 Mainz, Germany
[3]Centre of Natural Hazards and Disaster Science (CNDS), Uppsala University, Villavägen 16, 75236 Uppsala, Sweden
[4]Department of Meteorology, Stockholm University, Svante Arrhenius väg 16c, 114 18 Stockholm, Sweden
[5]Bolin Centre for Climate Research, Stockholm University, Svante Arrhenius väg 8, 114 18 Stockholm, Sweden

**Correspondence:** Antonio Segalini (antonio.segalini@geo.uu.se)

**Abstract.** The horizontal propagation of Rossby waves in the upper troposphere has been a long-standing topic in dynamical meteorology for a number of reasons. More recently, it has been suggested that the concept of "waveguidability" may be useful in this context. With an eye to such issues, the current paper suggests a novel and efficient algorithm to solve the linearized barotropic vorticity equation on the sphere in a forced-dissipative configuration. The algorithm allows one to obtain linear wave solutions resulting from arbitrary combinations of the forcing and the background zonal wind. Examples using single- and double-jet configurations illustrate that the proposed solutions allow one to quantify to what extent the background flow acts as a zonal waveguide for Rossby waves. The onset of barotropic instability might hinder the applicability of the linear framework, but it is shown that the nonlinear flow evolution can still be retrieved qualitatively from the linearized solution both for the stationary component of the wave field and for the evolution of transient waves.

## 1 Introduction

Rossby waves are a fundamental component of the upper-tropospheric dynamics (Rossby, 1939, 1940) and are instrumental in modulating the location, intensity, and track of extratropical weather systems. They can be forced by baroclinic instability, large-scale topographical features, and meso- to large-scale diabatic heating (e.g., Rhines, 1975; Hoskins and Karoly, 1981; Held, 1983; Garcia and Salby, 1987; Held et al., 2002; Brayshaw et al., 2009; Garfinkel et al., 2020; Martius et al., 2021). A key characteristic of Rossby waves, which has profound implications for their modulation of weather systems, is their propagation. The Earth's spherical geometry usually leads to the equatorward refraction of Rossby waves (Hoskins and Karoly, 1981), yet the presence of localised upper tropospheric jet streams can favour a zonal propagation (e.g., Hoskins and Valdes, 1990; Ambrizzi et al., 1995; Branstator, 2002; Wirth et al., 2018). The capability of jet streams to promote Rossby wave propagation, referred to as "waveguidability", has long been object of research (see the reviews by Wirth et al., 2018; White et al., 2022). Waveguides typically exist for time scales longer than the waves they "guide", and this property can be exploited to understand persistence and predictability of midlatitude weather (Martius et al., 2010).

The amplification and propagation of Rossby waves along such waveguides has been related in the literature to the occurrence of extreme weather events both in winter (e.g., Davies, 2015; Harnik et al., 2016) and summer (e.g., Kornhuber et al., 2019; Teng and Branstator, 2019; Di Capua et al., 2021; Rousi et al., 2022; Jiménez-Esteve et al., 2022; Kornhuber and Messori, 2023). In a number of these examples, Rossby waves were identified as quasi-stationary and associated with concurrent extremes in geographically remote regions due to the zonally-extended, large-amplitude nature of the waves. The existence of quasi-stationary waves on a mid-latitude waveguide has also been linked to the phenomenon of quasi-resonance, namely a constructive self-interference of the wave, which in turn may be linked to surface extreme events (Petoukhov et al., 2013; Coumou et al., 2014); however, whether or not quasi-resonance is a relevant mechanism in realistic situations is an unsolved question (Wirth and Polster, 2021).

The relevance that waveguidability plays in atmospheric dynamics, as well as its connection to surface extremes in a changing climate, has led to a renewed interest in its study (White et al., 2022). In general, waveguidability is regarded as a property of the background flow configuration in which Rossby waves are propagating. The concept of background flow is useful for decomposing the atmospheric flow into a background component that varies gradually in space and time, and a wave component (i.e., the Rossby waves in the present work) that varies more rapidly. The feasibility of such a decomposition inherently relies on the assumption that a clear scale separation exists between the waves and the background flow, such that the former are not leaving an immediate signature on the latter. The identification of the background flow can be practically done by using time and/or zonal averages, a methodology based on the assumed linearity of the waves: however, this assumption is violated when Rossby waves reach significant meridional amplitudes, as often happens in the context of large-scale flow configurations associated with unusual or extreme weather (Wirth and Polster, 2021).

The role of specific background flow configurations in promoting waveguidability can be investigated in detail using idealized frameworks. Manola et al. (2013) and Wirth (2020) are notable examples of such work for the cases of single, localized jet streams in the Northern Hemisphere. These studies noticed that waveguidability increases with the strength of the jet and that latitudinally narrow jets are more efficient waveguides than broad jets. Steady wave patterns have been discussed by means of the ray-tracing technique (Hoskins and Karoly, 1981; Hoskins and Ambrizzi, 1993; Wirth, 2020) where the $\beta$-plane analysis of Rossby waves is extended to slowly meridionally-varying background flows by means of WKB theory and a Mercator projection. This enabled to identify waveguides as ridges in the so-called refractive index, drastically simplifying the problem with respect to a full eigenvalue analysis. However, the ray-tracing approach is subject to some crucial limitations. It is often implemented on a Mercator projection of the flow field, leading to distortion in the high latitudes, and the meridional/longitudinal variation of the background flow must be gradual compared to the scale of the wave. Moreover, although ray-tracing can in principle be applied to transient waves, in practice it is typically used for the study of stationary waves (with the notable exception of Yang and Hoskins, 1996). One therefore loses information about how the waves are evolving in time or whether the flow will ever approach a steady state. A critical assessment of this approach was performed by Wirth (2020), albeit limited to only a few background flows. There is indeed a lack of understanding of Rossby waveguides in complex background flow configurations, such as the presence of two separate jet streams, which has been linked to summertime heatwaves ("double jets"; e.g., Coumou et al., 2014; Rousi et al., 2022). Understanding waveguidability thus requires a systematic investigation

of a large number of different background flows. Each case must also be investigated for different forcings to assess the flow response, and this task becomes quickly demanding in terms of computational resources and expert analysis time.

Motivated by this open challenge and by the limitations of ray-tracing theory, we extend in this paper the framework of Wirth (2020) to study the propagation of forced Rossby waves without recourse to ray-tracing and avoiding numerical integration of the underlying nonlinear equations. Similar approaches have been previously employed to study other types of oscillatory phenomena, such as Rossby wave critical layers (Campbell and Maslowe, 1998), equatorial waves (Boyd, 1978) and gravity waves (Baldauf and Brdar, 2013). Specifically, we study Rossby waves evolution by solving explicitly the linearized, 2-D barotropic vorticity equation in terms of normal mode analysis. This enables one to inexpensively obtain solutions for any forcing under a given zonally symmetric background flow; furthermore, it allows one to study Rossby wave propagation without the limitations imposed by ray-tracing methods, such as the assumption of scale-separation or the need to project the flow on a Mercator plane. We first validate the solutions by comparing them with the output of nonlinear numerical simulations based on a spectral code built on spherical harmonics. The joint analysis of linear and nonlinear solutions further allows a detailed investigation of the stability of different wave modes, providing information about the transient evolution of the Rossby waves. In the end we use the novel algorithm in order to extend the analysis of waveguidability of idealized zonal jets beyond what has been done in previous publications. The paper is structured as follows: section 2 presents the theory behind the linear approach, while its discrete implementation based on orthogonal Chebyshev polynomials is discussed in the Appendix A. A comparison between linear and nonlinear calculations is discussed in section 3 where different zonal wind profiles are investigated in terms of both the wave spatial structure and of integrated parameters such as the waveguidability. The stationary solutions for single-jet and double-jet configurations are discussed respectively in sections 4 and 5.3. The paper is closed by some concluding remarks in section 6.

## 2   Model framework and numerical details

Let us consider the two-dimensional barotropic vorticity equation on a spherical planet with radius $a^*$ (in this section dimensional quantities are indicated by means of an asterisk superscript)

$$\frac{\partial \zeta^*}{\partial t^*} + (\mathbf{V}^* \cdot \nabla_h)(\zeta^* + f^*) = -\chi^* \left(\zeta^* - \overline{\zeta}^*\right) + F^*, \tag{1}$$

where $\mathbf{V}^*$ is the horizontal velocity field, $f^* = 2\Omega^* \cos \theta$ is the Coriolis parameter, $\theta = \pi/2 - \varphi$ is the colatitude (associated with the latitude $\varphi$) and $\lambda$ is the longitude. Similarly to Hoskins and Ambrizzi (1993) and Wirth (2020), a damping term $-\chi^* \left(\zeta^* - \overline{\zeta}^*\right)$ is present: this term provides a stabilizing effect that hinders a too strong departure of the atmospheric state from the background vorticity, $\overline{\zeta}^*$. $F^*$ indicates a generic forcing in space and time. By normalizing physical quantities with respect to the planetary radius $a^*$ and a characteristic velocity $U_s^*$ such that

$$t = \frac{t^*}{a^*/U_s^*} \quad , \quad \mathbf{V} = \frac{\mathbf{V}^*}{U_s^*} \quad , \quad \zeta = \frac{\zeta^*}{U_s^*/a^*} \quad , \quad f = \frac{f^*}{2\Omega^*} = \cos \theta \quad , \quad F = \frac{F^*}{(U_s^*/a^*)^2} , \tag{2}$$

the barotropic vorticity equation (1) is re-written in dimensionless form as

$$\frac{\partial \zeta}{\partial t} + (\mathbf{V} \cdot \nabla_h) \left( \zeta + \frac{f}{\text{Ro}} \right) = -\chi \left( \zeta - \overline{\zeta} \right) + F \qquad \text{with} \qquad \text{Ro} = \frac{U_s^*}{2\Omega^* a^*}. \tag{3}$$

The choice of the characteristic velocity scale, $U_s^*$, is arbitrary but it should be of the same order of magnitude of the velocity field. In the considered two-dimensional case, the flow divergence is zero everywhere so that a streamfunction, $\Psi$, can be introduced, facilitating the determination of the velocity and vorticity field as

$$u_\theta = -\frac{1}{\sin\theta} \frac{\partial \Psi}{\partial \lambda} \quad , \qquad u_\lambda = \frac{\partial \Psi}{\partial \theta} \quad , \qquad \zeta = \nabla^2 \Psi. \tag{4}$$

By assuming a base zonal flow $u_\lambda = U(\theta)$ and $u_\theta = 0$, the undisturbed vorticity, $\overline{\zeta}$, is given by

$$\overline{\zeta}(\theta) = \frac{1}{\sin\theta} \frac{\partial}{\partial \theta} (U \sin\theta) = \cot\theta\, U + \frac{\partial U}{\partial \theta} \qquad \text{for} \qquad 0 < \theta < \pi. \tag{5}$$

Let us consider a perturbed state where the relative vorticity is given by $\zeta = \overline{\zeta} + \zeta'$. This will be associated with a stream-function, $\overline{\Psi} + \psi'$, and velocity field

$$\mathbf{V} = (u_\theta, u_\lambda) = (0, U) + (u_\theta', u_\lambda') = (0, U) + \left( -\frac{1}{\sin\theta} \frac{\partial \psi'}{\partial \lambda}, \frac{\partial \psi'}{\partial \theta} \right). \tag{6}$$

The equation governing the small perturbation is obtained by taking (3) and subtracting the base-state equation. The forcing term is not included in the background flow equation and generates a deviation from the latter. This deviation is accounted for as a perturbation. Furthermore, wave-mean flow terms are not accounted for, implying that the perturbation flow is assumed not to modify the background flow itself. By also neglecting the nonlinear wave-wave terms, one obtains the linearised barotropic vorticity equation

$$\frac{\partial \zeta'}{\partial t} + \frac{U}{\sin\theta} \frac{\partial \zeta'}{\partial \lambda} + u_\theta' \frac{\partial}{\partial \theta} \left( \overline{\zeta} + \frac{f}{\text{Ro}} \right) = -\chi \zeta' + F, \tag{7}$$

with

$$\zeta' = \nabla_h^2 \psi' \quad , \qquad u_\theta' = -\frac{1}{\sin\theta} \frac{\partial \psi'}{\partial \lambda}. \tag{8}$$

The system can be better analysed by taking the Fourier transform in the zonal direction defined as

$$\widehat{\zeta}(\theta; m) = \int_0^{2\pi} \zeta(\theta, \lambda)\, e^{-im\lambda}\, \mathrm{d}\lambda, \tag{9}$$

leading to

$$\frac{\partial \widehat{\zeta}}{\partial t} + \left( \frac{imU}{\sin\theta} + \chi \right) \widehat{\zeta} + \widehat{u_\theta} \frac{\partial}{\partial \theta} \left( \overline{\zeta} + \frac{f}{\text{Ro}} \right) = \widehat{F}, \tag{10}$$

where

$$\widehat{\zeta} = \frac{\partial^2 \widehat{\psi}}{\partial \theta^2} + \cot\theta \frac{\partial \widehat{\psi}}{\partial \theta} - \frac{m^2}{\sin^2\theta} \widehat{\psi} = \mathcal{L}\widehat{\psi} \quad , \qquad \widehat{u_\theta} = -\frac{im}{\sin\theta} \widehat{\psi}, \tag{11}$$

and $\mathcal{L}$ indicates the Laplace operator that, once discretised, becomes a matrix. By introducing the matrix $B = i\mathcal{L}$ and the streamfunction vector $\widehat{\psi}_j = \widehat{\psi}(\theta_j; m)$ evaluated at the colatitudes $\theta_j$, the barotropic vorticity equation (10) is written as

$$-iB\frac{\partial \widehat{\psi}_j}{\partial t} + \left[\left(\frac{imU}{\sin\theta} + \chi\right)\mathcal{L} - \frac{im}{\sin\theta}\frac{\partial}{\partial\theta}\left(\overline{\zeta} + \frac{f}{\text{Ro}}\right)I\right]\widehat{\psi}_j = -iB\frac{\partial \widehat{\psi}_j}{\partial t} + A\widehat{\psi}_j = \widehat{F}, \qquad (12)$$

where the terms composing the matrix $A$ are grouped within the square brackets in (12) and $I$ is the identity matrix (see Appendix A for a detailed description of how the differential operators are discretized and the matrices are constructed). Equation (12) is a linear system of ordinary differential equations in time with the streamfunction at the collocation points as unknowns. By erasing the time derivative, the equilibrium state of the linearised system is obtained as

$$\widehat{\psi}_j = A^{-1}\widehat{F}, \qquad (13)$$

similar to the formula provided by Hoskins and Karoly (1981). Eq. (13) corresponds to the steady solution of the forced dynamical system achieved after a sufficiently long time if the system is stable (see discussion below for the flow stability analysis). Even if the system is unstable, the state described by Eq. (13) is the only one that erases the time derivative of the linearised system: thus, we choose to name it "equilibrium solution". Its determination is only related to the advection and dissipation of vorticity rather than the vorticity temporal evolution and can therefore be obtained in one computational step.

The solution of Eq. (12) describing the growth of an infinitesimally small perturbation (be it initiated by the forcing or by a generic initial condition) can be written as the sum of the homogeneous solution (starting from a given initial condition) plus a forced solution. By introducing a perturbation as a modal ansatz $\widehat{\psi}_j = \widetilde{\psi}_j e^{-i\omega t}$ it is possible to solve the homogeneous problem as an eigenvalue one, identifying the complex eigenvalues, $\omega$, and the associated eigenfunctions for each azimuthal wavenumber, $m$. The advantage of the proposed framework is that the behaviour of the system is determined from the eigen-
values of the homogeneous problem, that are *independent* of the forcing and have indeed general validity for a given zonal background flow. If all the complex eigenvalues have negative imaginary part the system is stable and it will converge to the equilibrium state given by Eq. (13), while the perturbations will grow with time if at least one eigenvalue has positive imaginary part. Thus, the sign of the imaginary part of the eigenvalues determines whether a perturbation will grow or decay. The real part of the eigenvalues, on the other hand, is associated with phase propagation in the zonal direction. In this framework,
the evolution of the perturbation is obtained as an initial value problem, drastically simplified by means of the modal analysis (details are provided in the Appendix A). As a note of caution, the linearisation is performed around the background state, $\overline{\Psi}$, while the presence of the forcing shifts the equilibrium state of the system to $\overline{\Psi} + \mathcal{F}^{-1}\left(A^{-1}\widehat{F}\right)$ (with $\mathcal{F}^{-1}$ indicating the inverse zonal Fourier transform). The modal analysis assumes infinitesimal deviations from the equilibrium state, while here it is performed around the background flow. The two linearizations are identical when the forcing is infinitesimal, but the modal
analysis is still expected to be qualitatively similar for reasonably small forcing, an hypothesis to be verified a posteriori. The choice to linearize around the background flow has the practical advantage that the Fourier decomposition can be applied to the linearized barotropic vorticity equation without the appearance of convolution terms that would hinder the separation of individual waves. Furthermore, as the forcing is not included in the modal analysis, its contribution is only later observed into the waves, making the computational approach simpler and more insightful.

## 3 Model setup and validation

The proposed linearized framework, based on Chebyshev polynomials for the treatment of the solution in the meridional direction, is first compared with the test cases proposed by Wirth (2020) for two of the investigated zonal velocity profiles. The code has been written in Python, where Chebyshev polynomials have been implemented following Peyret (2002) and Canuto et al. (2006). In all simulations, only the dissipative term $-\chi^* \nabla_h^2 \psi'$ (with $\chi^* = (7 \text{ d})^{-1}$ as in Wirth, 2020) was used to make the barotropic wave decay and no hyperviscosity was introduced. The planetary radius was taken as 6371 km while the planetary rotation speed was $\Omega = 7.292115 \cdot 10^{-5} \text{ rad s}^{-1}$. The linear problem has been discretized into an equal number of latitudes and longitudes without any aliasing consideration or numerical stability constraints (see Appendix A for the numerical details).

In order to check the code correctness, an independent numerical solution of the barotropic vorticity equation has been implemented by using the spherical harmonics transform (SHT) package from Schaeffer (2013). Both the linear and nonlinear barotropic vorticity equations were implemented and compared to the results of the proposed linearised framework. A triangular truncation scheme was adopted with an aliasing-removal approach in both the linear and nonlinear simulation with spherical harmonics (although this made a difference only for the nonlinear simulations). A leapfrog scheme was used for the temporal discretization with a time step of 10 minutes. A Robert-Asselin filter with filter parameter 0.01 was implemented to eliminate the spurious computational mode associated with the leapfrog method (Kalnay, 2003).

Since the topographic forcing $F$ is stationary, the constant forcing solution provided by Eq. (A7) is used. The expression is further simplified here as the initial perturbation streamfunction is assumed to be zero, implying the vanishing of the first term of Eq. (A7), so that only the forced response needs to be computed. The forced response is obtained from the method of variation of constants and therefore facilitated by the eigenfunctions calculated from the homogeneous problem

In the special case of solid body zonal velocity profile, $U = U_0 \sin \theta = U_0 \cos \varphi$, the zonal velocity is obtained as if the atmosphere was rotating at a slightly higher angular velocity than $\Omega$ and the linearised homogeneous barotropic vorticity equation (7) reduces to

$$\frac{\partial \nabla_h^2 \psi'}{\partial t} + U_0 \frac{\partial \nabla_h^2 \psi'}{\partial \lambda} + 2 \left( U_0 + \Omega a \right) \frac{\partial \psi'}{\partial \lambda} + \chi \nabla_h^2 \psi' = 0. \tag{14}$$

Equation (14) can be solved analytically by means of the ansatz $\psi' \propto Y_l^m (\theta, \lambda) e^{-i\omega t}$ where $Y_l^m$ is the spherical harmonic with degree $l$ and order $m$. The resulting dispersion relationship is

$$\omega = m \left[ U_0 - \frac{2U_0 + 2\Omega a}{l(1+l)} \right] - i\chi, \tag{15}$$

which is a well-known analytical result for Rossby-Haurwitz waves (Haurwitz, 1940). The attenuation parameter $\chi$ makes the eigenvalue stable, while the first term of Eq. (15) is associated with the zonal phase propagation of the wave, similarly to the planar case. From this analysis it is already known that: (i) all the modes of the solid-body zonal velocity case are stable (since $\chi > 0$), (ii) the modes are spherical harmonics and (iii) the dispersion relationship is given by Eq. (15). Simulations done with different grid resolutions showed an excellent agreement between the analytical relationship (15) and the numerical eigenvalues, with error comparable to the precision of the machine (not shown).

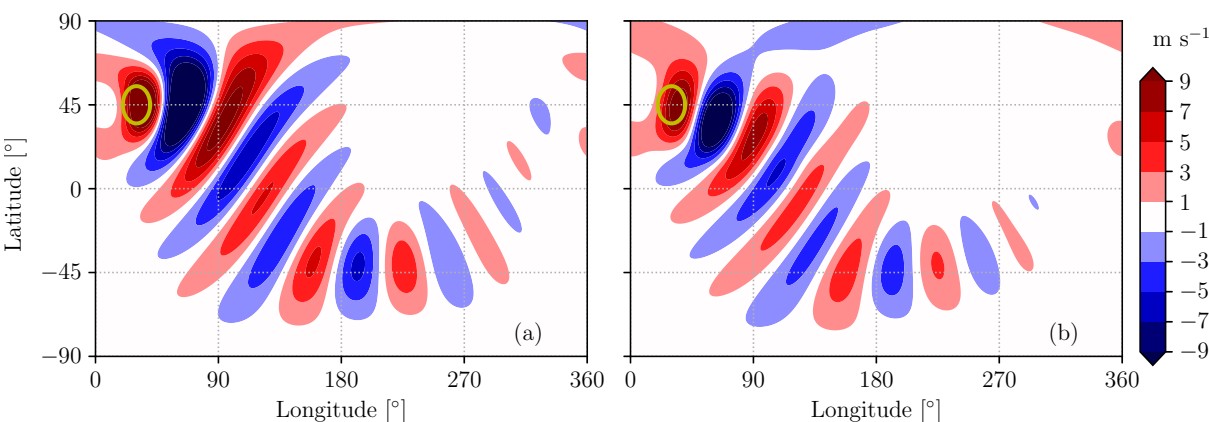

**Figure 1.** Meridional velocity pattern at the equilibrium state for the zonal velocity profile $U = 15 \cos\varphi \,\mathrm{m\,s^{-1}}$ from the linear analysis (a) and after 100 days with the nonlinear solver (b). The ellipse indicates the topographic forcing.

Fig. 1a shows the meridional velocity field (positive northward) and the associated wave pattern at the equilibrium state (13) forced by a smooth, idealized mountain located at latitude $\varphi_F = 45°\,\mathrm{N}$ and longitude $\lambda_F = 30°\mathrm{E}$ and described by

$$180 \quad F = -7.73 \cdot 10^{-9} \left(\lambda - \lambda_F\right) \exp\left[-\frac{(\varphi - \varphi_F)^2}{2\sigma_{\varphi,F}^2} - \frac{(\lambda - \lambda_F)^2}{2\sigma_{\lambda,F}^2}\right] h_F \,, \tag{16}$$

where $\sigma_{\varphi,F} = \sigma_{\lambda,F} = 10°$ and $h_F = 0.3$, unless otherwise stated. The zonal background flow is given by $U = 15 \cos\varphi \,\mathrm{m\,s^{-1}}$ (or with $U_0 = 15 \,\mathrm{m\,s^{-1}}$ as above). Fig. 1a should be compared to the nonlinear solution in Fig. 3a of Wirth (2020): the structure of the wave pattern is very similar, although here obtained without any time integration since the flow will in time approach the steady-state solution because the imaginary part of the eigenvalues is negative. The linear spherical harmonics solution is practically identical to the Chebyshev approach and therefore the results will not be shown here. The nonlinear solver has slight differences from the linear method, such as a more rapid wave attenuation away from the forcing (as visible in Fig. 1b).

A sensitivity study can be performed for different grid resolutions to assess the appropriate number of polynomials to be used in order to achieve a desired convergence. The waveguidability, calculated as defined in Wirth (2020), has been used here as the key quantity for the comparison. The estimate by Wirth (2020) corresponds to the ratio between the enstrophy found in a monitoring region downstream of the forcing (180° E and 270° E, 30° N and 60° N) and the total enstrophy integrated over all latitudes across 180° E and 270° E. Fig. 2a shows the convergence of the waveguidability metric around 45° N for different grid resolutions in order to quantify the minimum number of polynomials needed for numerical convergence. It is noteworthy that the convergence of the results is obtained even for moderate resolutions with the linear Chebyshev and linear SHT methods, while the nonlinear method approaches a higher value of the waveguidability metric, suggesting a difference between linear and nonlinear analyses. Having assessed the grid convergence, the rest of the work will use grid resolution $N = 256$ for the solution of the linearized eigenvalue problem via Chebyshev polynomials while a T170 truncation will be used for the nonlinear spherical harmonics simulations. Since the linear Chebyshev and the linear SHT code agree with each other, and are both based on Eq. (7), only the Chebyshev implementation will be discussed in the following and it will be

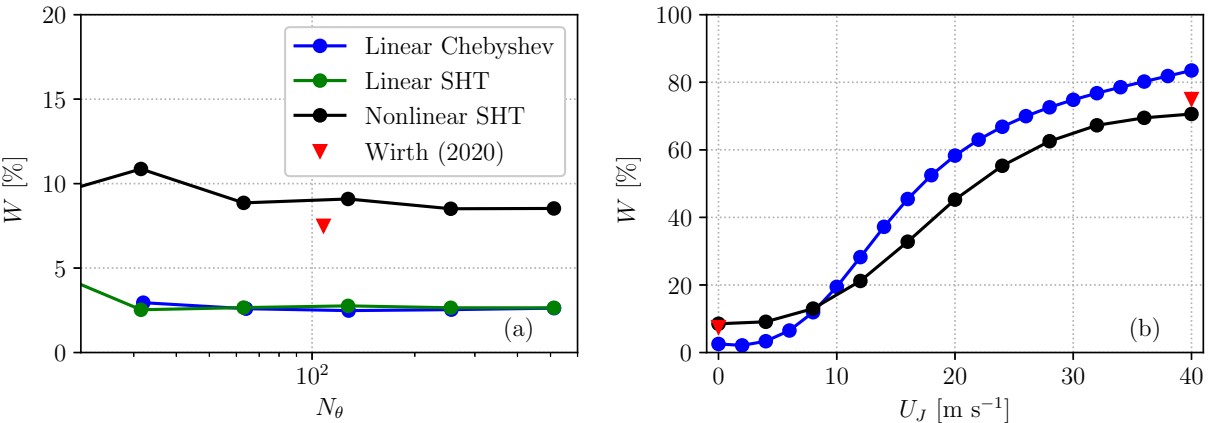

**Figure 2.** (a) Waveguidability for a Gaussian mountain located at $\varphi_F = 45°$ N estimated from different linear and nonlinear simulations at different grid resolutions when $U = 15\cos\varphi$ m s$^{-1}$ (namely without any latitudinally-confined jet). $N_\theta$ indicates the number of latitude grid points. (b) Waveguidability assessed for the linear and nonlinear simulations for $N = 256$ (corresponding to a T170 resolution) for different jet velocities $U_J$ (according to the zonal velocity profile given in Eq. 17).

referred to as the *linear* simulation. The only nonlinear implementation of Eq. (3) is by means of the SHT, which will be hereafter denoted simply as *nonlinear* simulation.

## 4 Single-jet configuration: stability and dynamics

We will now focus on the case of a latitudinally-confined jet described by

$$U = U_0 \cos\varphi + U_J \exp\left[-\frac{(\varphi - \varphi_J)^2}{2\sigma_J^2}\right] + L(\varphi), \tag{17}$$

where $U_J$ is the jet velocity, $U_0 = 15$ m s$^{-1}$, $\varphi_J = 45°$ N is the jet latitude and $\sigma_J = 5°$ (unless otherwise stated). $L$ indicates a linear correction that imposes $U = 0$ at the two poles.

With this setup, we notice that the linear method is capable of replicating the waveguidability increase with jet speed, obtaining similar values as Wirth (2020) for a jet located at 45° N (Fig. 2b). Interestingly, the linear method features a convergence of the solution only for not-too-high jet speeds: in particular, for a speed faster than 20 m s$^{-1}$, some of the eigenvalues $\lambda_i$ acquire a positive imaginary part and lead to divergence. The reasons for this behavior, and the usefulness of the linear method even in such a case, will be illustrated in Sec. 4.1 and 4.2.

The difference in waveguidability metric between the linear and nonlinear solutions must be related to a dissipative effect of the nonlinear terms (accounting for the effect of the waves on other waves and the basic state), neglected in the linear analysis. The above waveguidability metric, however, cannot be easily applied to compare waveguides located at different latitudes: in fact, the physical distance between the forcing and the monitoring sector would vary when the jet latitude is changed, leading

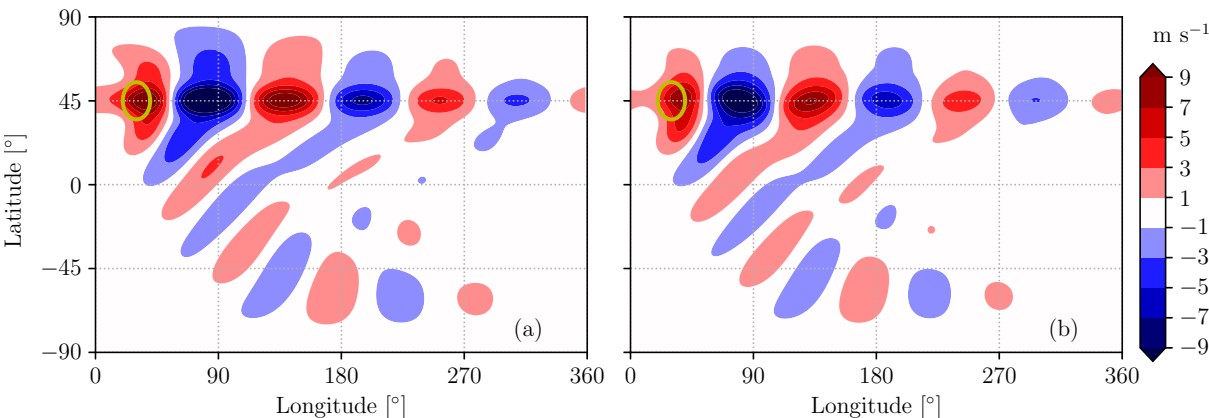

**Figure 3.** Meridional velocity in the strong-jet case ($U_J = 40 \, \mathrm{m \, s^{-1}}$) from the linear method at the equilibrium state (a). Meridional velocity field averaged between 10 and 100 days from the simulation start with the nonlinear solver (b). The ellipse indicates the topographic forcing.

to a spurious increase of waveguidability towards the Pole. An adapted metric will be proposed in Sec. 5.1 in the attempt to provide a definition able to account for jets at different latitudes.

## 4.1 Stability analysis

As noticed by Wirth (2020), the strong-jet case with $U_J = 40 \, \mathrm{m \, s^{-1}}$ is characterized by an unsteady velocity field that does not achieve a steady state, regardless of the integration time. This unsteady behavior is partly explained by the eigenvalues provided by the linear analysis: some eigenvalues have in fact positive imaginary part and, therefore, the flow field is unstable. According to linear theory, the atmospheric state should diverge exponentially from the equilibrium state and each unstable Rossby wave should grow without bounds. The nonlinear simulation, however, does not display such an unrealistic divergence of wave amplitudes. A great-circle wave propagation is still present as in the solid-body velocity case, as well as a wave pattern in the zonal direction of the waveguide corresponding to the strong jet (Fig. 3a).

We conduct here a systematic stability analysis to identify at which jet velocity and width unsteady traveling-wave patterns begin to appear. The Rayleigh stability criterion provides a necessary condition for the onset of barotropic instability, namely a change in sign of the absolute vorticity gradient $\partial(\bar{\zeta} + 2\Omega a \cos\theta)/\partial\theta$. The derivation of the criterion can be applied to Eq. (10) leading to the result that, whenever the absolute vorticity gradient changes sign, it is allowed that the imaginary part of $\omega$ is different from $-\chi$, with a significant stability margin before the actual onset of the linear instability (namely when the imaginary part of $\omega$ becomes positive). This results pinpoints the role of dissipation in retarding the onset of barotropic instability, confirming that the Rayleigh criterion provides a necessary but not sufficient condition for the onset of instability (Kuo, 1949).

Let us consider first the results of the eigenvalue analysis for a narrow jet zonal profile ($\sigma_J = 5°$; blue dots in Fig. 4) over an array of different jet speed $U_J$ and latitude $\varphi_J$. At low $U_J$ the imaginary part of the most unstable eigenvalue is determined by the dissipation parameter $\chi$. This is valid as long as the absolute vorticity gradient has the same sign. As $U_J$ increases, the

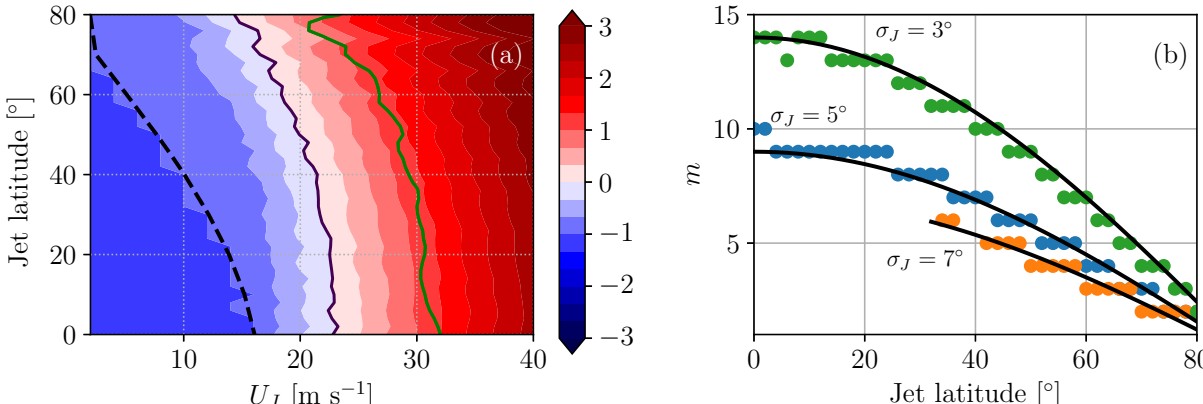

**Figure 4.** (a) Maximum of the imaginary part of the linear eigenvalues for a given jet velocity (normalised by $\chi$) when $\sigma_J = 5°$. The black line indicates the neutral curve, the dashed line is the locus where the absolute vorticity gradient changes sign (Rayleigh stability criterion), while the green thick line is the locus beyond which the temporal variance of the meridional velocity in the nonlinear simulation becomes larger than $2\,\mathrm{m^2\,s^{-2}}$ (equivalent to the neutral curve in the nonlinear case). (b) Azimuthal wavenumber of the most unstable eigenvalue for $U_J = 40\,\mathrm{m\,s^{-1}}$ for different jet widths and latitudes. The black lines are curves $m \propto \cos\varphi_J$ fitting the linear stability results.

stability margin decreases: above 15-22 $\mathrm{m\,s^{-1}}$ at least one eigenvalue becomes positive, leading to the divergence of the linear solution. Nonlinear simulations follow a similar pattern, but the solution approaches an unstable state for larger jet velocity than for the linear method (as indicated by the green line in figure 4a denoting the locus where the meridional velocity time variance equals $2\,\mathrm{m^2\,s^{-2}}$). Above this threshold, the velocity variance increases drastically and it is concentrated at the jet location

irrespective of the forcing. The neutral curve from the linear stability analysis is then a conservative estimate of the onset of instability, when compared to the onset of unsteady fluctuations in the nonlinear analysis. This underlines the stabilizing role of nonlinear terms not included in the linear method, such as wave-wave interaction. A weaker jet velocity is needed for the instability onset when the jet approaches the pole, where the linear and nonlinear neutral curves are closer to each other.

It is also interesting to monitor the wavenumber associated with the most unstable eigenvalue. Beyond the onset of the linear

instability, this wavenumber does not depend on $U_J$, and therefore the wavenumber associated to the maximum growth rate in Fig. 4b is only shown for $U_J = 40\,\mathrm{m\,s^{-1}}$, where the jet is unstable at all latitudes. For $\varphi_J = 45°$ N a wavenumber $m = 6$ is the most unstable, the wavenumber typical of midlatitude Rossby waves. However, as the jet is shifted to the pole, the most unstable wavenumber systematically decreases following the cosine of $\varphi_J$. It is possible to explain this trend by assuming that the instability is associated with a wavelength independent of the latitude, given by

$$L = \frac{2\pi}{m} a \cos\varphi = \text{constant}\,, \tag{18}$$

which is indeed proportional to $\cos\varphi$. This implies that the most barotropically-unstable Rossby wave has the same wavelength at different latitudes of the zonal jet.

Inspired by the analysis of Gill (1982), it is expected that the Rossby wave will become more unstable for narrower jet width, $\sigma_J$, and conversely less unstable for a wider jet. Fig. 4b shows indeed the zonal wavenumber of most unstable mode for

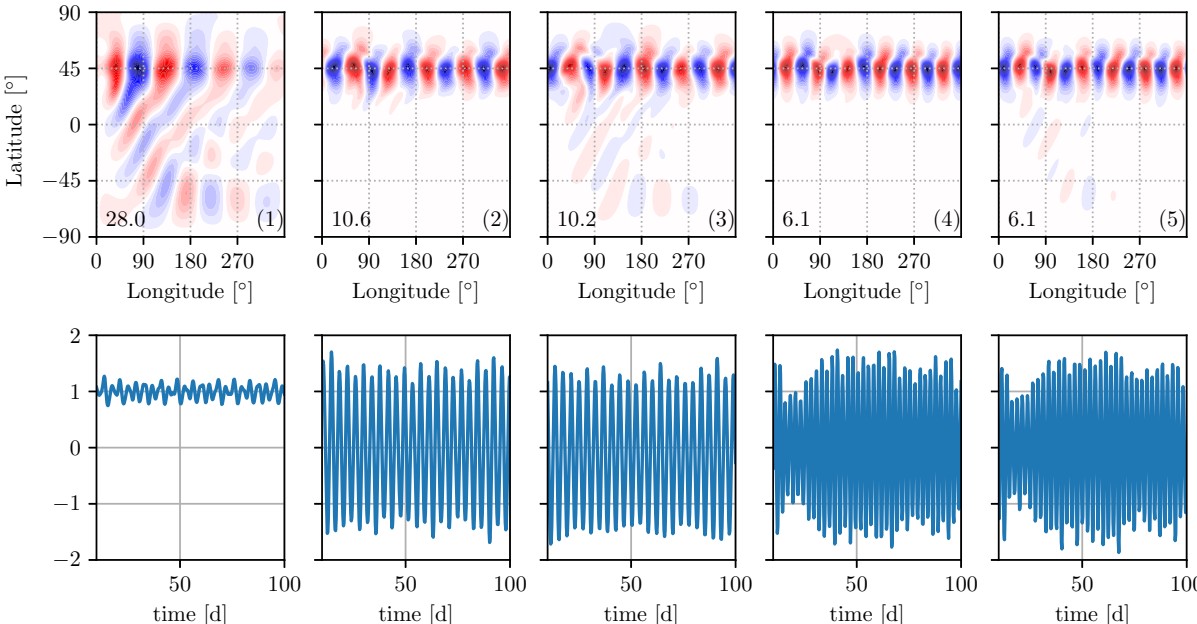

**Figure 5.** First five modes of the EOF decomposition of the meridional velocity field obtained from the nonlinear simulations (top) and their temporal coefficient normalized by their root mean square value (bottom) for the strong-jet case with $U_J = 40 \ \mathrm{m\,s^{-1}}$. The number in the bottom-left corner of the first row indicates the variance contribution of each mode. The color scale in the top row is not consistent between the different modes since each mode has unitary norm.

$U_J = 40 \ \mathrm{m\,s^{-1}}$ and three different jet widths ($\sigma_J = 3°$, $5°$ and $7°$). In general, modes associated with a narrow jet exhibit a higher wavenumber than for a broad jet. Interestingly, the broadest jet ($\sigma_J = 7°$) is unstable only when the jet is far enough from the equator and it is stable for $\varphi_J < 30°$ N. By having the even higher value $\sigma_J = 10°$, the instability disappears at all jet latitudes and the flow becomes stable again at $U_J = 40 \ \mathrm{m\,s^{-1}}$ (not shown).

### 4.2 Dynamics of the unstable modes

After having mapped the conditions that determine the onset of instability, and related them to the onset of barotropic instability, we describe here the type of instability present in the model at high jet speeds. The linear eigenvalue analysis points out that the equilibrium state given by Eq. (13) is indeed unstable. Nevertheless, such an unstable equilibrium state resembles quite closely the time-averaged nonlinear solution, which is bounded and characterized by the averaged field reported in Fig. 3b. Beyond the linearly stable range, only the nonlinear simulations can provide information about the flow evolution and whether or not an equilibrium state is achieved. In order to shed some light, an empirical orthogonal function (EOF) approach has been applied to the nonlinear simulation for a single jet with $U_J = 40 \ \mathrm{m\,s^{-1}}$ and $\varphi_J = 45°$ N. The EOF algorithm was applied on the meridional velocity field between 10 days and 100 days after the start of the simulation without removing the time average. This provided a set of modes orthogonal to each other, including the time averaged one, which facilitates the

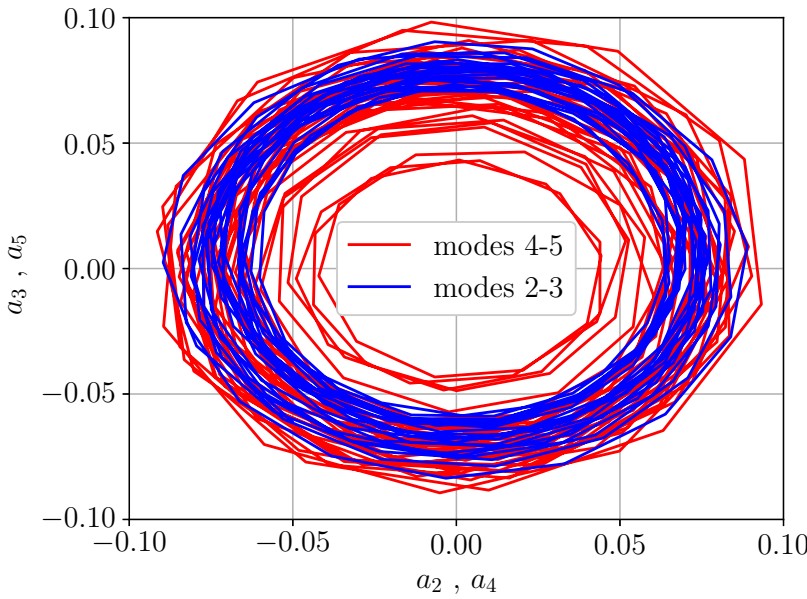

**Figure 6.** Evolution of the temporal coefficient of mode 2 (4) plotted against the one of mode 3 (5) from the EOF analysis shown in Fig. 5. The temporal coefficients have been normalised by their root mean square.

development of a reduced-order model. The corresponding modes are shown in Fig. 5 to highlight the most energetic scales governing the temporal flow evolution. The first mode is practically constant in time and remains close to the time-averaged field of the nonlinear simulation, while the second-third, fourth-fifth, and sixth-seventh modes organize themselves to create traveling wave patterns with zonal wavenumbers $m = 5$, $m = 6$ and $m = 7$, respectively (the sixth-seventh modes are not shown). These first seven modes contain 60% of the meridional velocity variance. The modal coefficients of mode 2 and 3 are in quadrature with each other, and similarly happens for modes 4 and 5. The trajectories in the plane spanned by the corresponding temporal coefficients, shown in Fig. 6, belong to a closed orbit and behave as traveling waves: however, the linear stability analysis does not support this interpretation, as the waves should grow exponentially in magnitude because they are unstable. A plausible alternative is that a limit cycle takes place (Strogatz, 2018): since a limit cycle is associated with a nonlinearity of the system, a linear analysis cannot identify that, and this is consistent with the divergence of the linear solution for small enough perturbations. The limit-cycle interpretation is supported by additional nonlinear simulations discussed in Appendix B. The results of those simulations indicate that the linear instability mechanism amplifies the unstable waves which, however, cannot grow indefinitely and are damped by wave-wave interactions when their amplitude becomes too large. When the linear growth balances the nonlinear attenuation the wave becomes stable and behaves like a travelling wave.

Additional simulations where the orographic forcing was moved to lie right on the equator did not change this overall picture, and the traveling waves (i.e., the EOF modes beyond the mean component) remained at the jet latitude (not shown). This implies that the wave components described by the EOF analysis are traveling waves determined just by the background flow, rather than by the forcing, similarly to what suggested by the linear analysis. One might wonder if there is a correspondence between

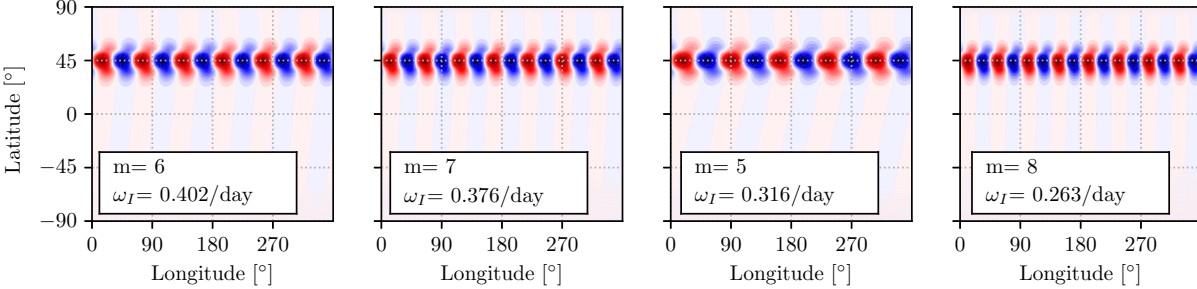

**Figure 7.** The 4 most unstable modes from the linear analysis sorted according to their growth rate in the strong-jet case.

the unstable modes of the linear analysis and the most energetic EOF modes. The linear unstable waves are located at the jet latitude (Fig. 7) and have similar wavenumbers to the first EOF modes. Despite the similarity, however, the most-rapidly growing mode ($m = 6$, Fig. 7) does not exactly match the EOF patterns corresponding to the modes 2 and 3 (which appear to project on $m = 5$, cf. Fig. 5).

## 5 Waveguidability assessment

### 5.1 Extending the Wirth (2020) diagnostic

Determining a universal estimate of waveguidability is still an open question. A universal waveguidability metric should inform about the ability of waveguides to duct waviness and should be in principle applicable to any background flow. While developing such a metric metric is not the focus of the current study, we nonetheless need a metric that enables the comparison of background flows with jets at different latitudes. Here, we opt to consider the amount of enstrophy that remains at the forcing latitude compared to the total one. This implies a minimal change with respect to Wirth (2020), with an integration over all longitudes rather than just a sector. The normalized meridional enstrophy density $E$ of a layer centered around $\varphi = \varphi_0$ is here defined as

$$E(U_J, \varphi_0) = \frac{\int_{\varphi_0 - \Delta\varphi}^{\varphi_0 + \Delta\varphi} \cos\varphi \int_0^{2\pi} \mathcal{E}(\varphi, \lambda)\, \mathrm{d}\lambda\, \mathrm{d}\varphi}{\int_{-\pi/2}^{\pi/2} \cos\varphi \int_0^{2\pi} \mathcal{E}(\varphi, \lambda)\, \mathrm{d}\lambda\, \mathrm{d}\varphi}, \tag{19}$$

where $\Delta\varphi = 15°$ and $\mathcal{E}(\varphi, \lambda) = \langle \zeta - \overline{\zeta} \rangle^2/2$ is the enstrophy of the 90 days time averaged vorticity anomaly from the background state (if a temporal steady state is achieved, the time averaging is unnecessary and only the final time is considered). The time averaging operation is indicated by $\langle \cdot \rangle$. It is important to consider that this definition requires a forcing centered around the same latitude where $E$ is computed, $\varphi_0$, as a source of vorticity. The so-defined normalized meridional enstrophy density takes into account the enstrophy in the vicinity of the forcing, which will be high even in absence of jet streams and thus cannot be directly used as a measure of the waveguidability. Within the definition (19) the jet latitude, $\varphi_J$, forcing latitude, $\varphi_F$, and latitude where the enstrophy anomaly is computed, $\varphi_0$, coincide to the same value when assessing how strong a jet

stream is: this enables the evaluation of the jet's capacity to hold the enstrophy (injected at the same latitude of the jet stream) within itself.

## 5.2 The single-jet case

With the help of the newly developed linear method, it is possible to efficiently estimate the equilibrium state for many jet velocities and jet latitudes and use it to calculate $E$ for the linear simulations. This is shown in Fig. 8a for single jets with different jet latitudes and strengths. It is clear that the stronger the jet, the higher the normalized enstrophy density $E$, as noticed already for the jet at 45° N. However, when no jet is present (and therefore no waveguide should exist) $E$ remains still high due to the enstrophy generated in the vicinity of the forcing. We also notice that $E$ increases with latitude, as more enstrophy is found along the short latitude circles close to the Pole.

In order to isolate just the contribution of the waveguide in keeping the energy at that latitude, Fig. 8b shows the difference $\Delta E$ between $E(U_J, \varphi_0)$ and $E(0, \varphi_0)$ (namely the normalized enstrophy density without any jet). Removing the contribution of solid-body rotation isolates the increase in $E$ due to the presence of a waveguide. We see that $\Delta E$ increases with jet speed in a consistent way for all latitudes. An equatorial westerly jet (which is obviously not relevant to realistic situations) results in a weaker $\Delta E$ than in midlatitudes; this last result is presumably due to the fact that the Equator already constitutes a waveguide for a background solid body rotation even without any jet superimposed.

The increment $\Delta E$ due to the presence of the jet is, as expected, small for low jet speeds. However, even for the high jet speed of 40 m s$^{-1}$ $\Delta E$ remains below 50% suggesting that $\Delta E$ is not an appropriate waveguidability metric for strong jets. In order to propose a simple assessment of the waveguidability property, one could consider how $\Delta E$ compares to an "ideal" waveguide at the same latitude, i.e., a waveguide that would keep all the enstrophy in the same latitude circle of the jet ($E = 1$). This could be computed as the ratio

$$W(U_J, \varphi_0) = \frac{E(U_J, \varphi_0) - E(0, \varphi_0)}{1 - E(0, \varphi_0)}, \tag{20}$$

that is defined to be a number between 0 (no jet) and 1 (ideal waveguide). This ratio gives a direct measure of how the normalized enstrophy density is increased by the jet presence, relative to the increase achievable with an ideal waveguide. The so-defined waveguidability metric (20) increases with jet speed from 0 to around 90%, as expected, but it also features an increase with latitude for strong jets (Fig. 8c). This trend of $W$ with latitude is likely not due to the fact that high-latitude jets act as better waveguides than low-latitude jets, because $\Delta E$ (i.e., the numerator of $W$) does not feature an analogous latitudinal variation. On the other hand, it likely results from the relative variation of $E(0, \varphi_0)$ with latitude visible in Fig. 8a, which leads to a reduction of the denominator of Eq. (20) at high latitudes. From a physical point of view, this means that a large part of the enstrophy is already kept at the latitude of the forcing even in the absence of a jet: For this reason, it is much easier to approach the status of "ideal waveguide" at high latitudes than at low latitudes, at least according to our metric.

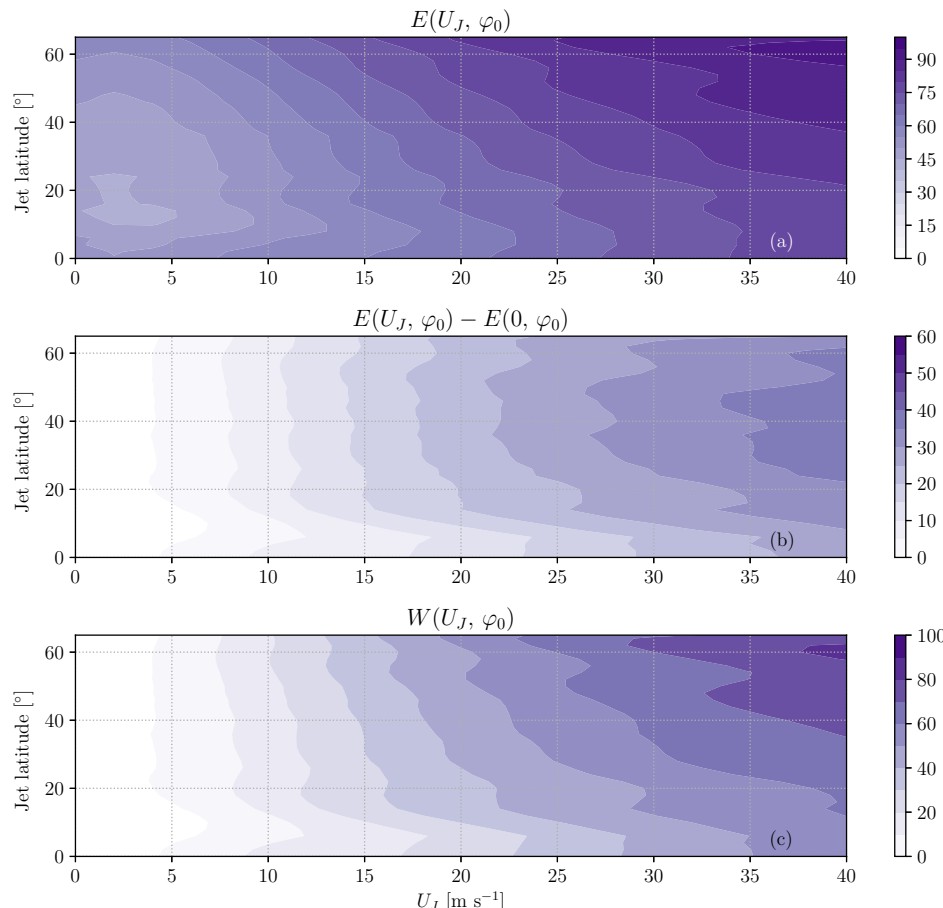

**Figure 8.** (a) Normalized meridional enstrophy density of the single jet zonal profiles (as in Eq. 17) for different jet latitudes and strengths, estimated according to Eq. (19) with $\varphi_0$ equal to the jet latitude. The enstrophy field has been computed from the equilibrium state obtained from the linear method. Only narrow jets with $\sigma_J = 5°$ have been considered. (b) Difference $E(U_J, \varphi_0) - E(0, \varphi_0)$ used to highlight the increment in normalized enstrophy density with the jet speed with respect to the solid-body case. (c) Estimated waveguidability calculated according to Eq. (20).

## 5.3 The double-jet case

After having considered the single-jet case, we now consider a configuration with two separate jets of width $\sigma_J = 5°$ located
at different latitudes. The double-jet setup was chosen as it is representative of the interplay between the subtropical and the eddy-driven jet streams observed in the Northern Hemisphere, and because recent research has connected it to the occurrence of quasi-stationary Rossby waves and summer heat extremes (Rousi et al., 2022). The first jet is located at $\varphi_J = 30°$ N and has a jet velocity of $U_J = 40 \, \mathrm{m \, s^{-1}}$ (the associated perturbation field is shown in Fig. 9 once again for both the linear and nonlinear

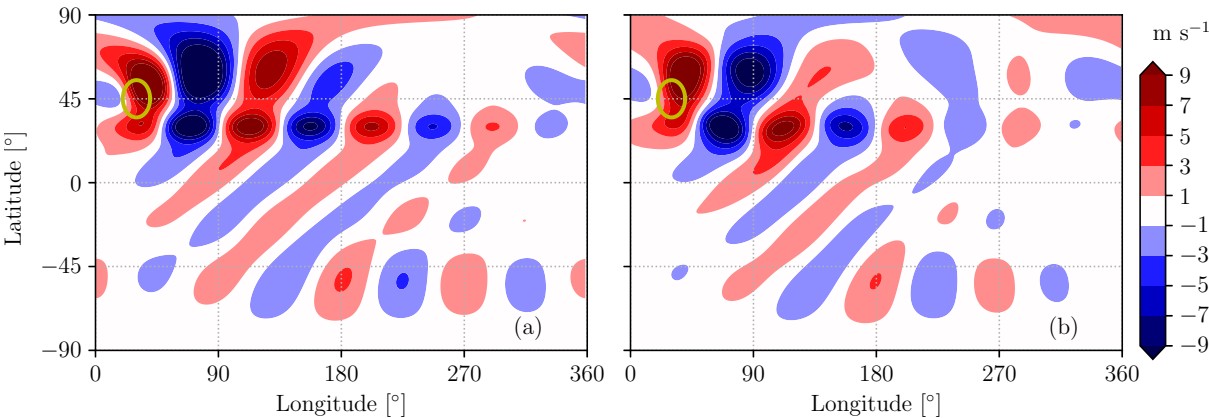

**Figure 9.** Meridional velocity associated with a zonal jet characterized by $U_J = 40 \ \mathrm{m\,s^{-1}}$ with $\varphi_J = 30°$ N at the equilibrium state from the linear analysis (a) and averaged between 10 and 100 days from the simulation start with the nonlinear solver (b).

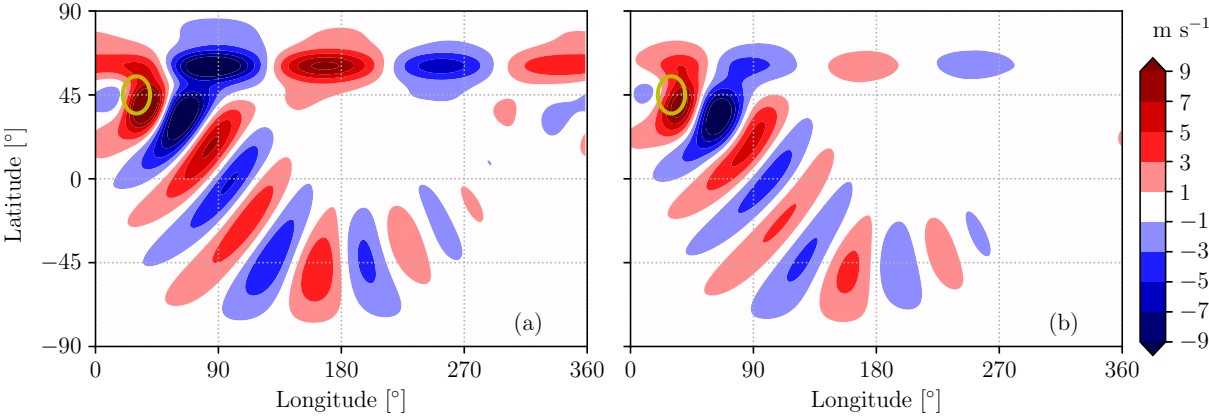

**Figure 10.** Meridional velocity associated with a zonal jet characterized by $U_J = 40 \ \mathrm{m\,s^{-1}}$ with $\varphi_J = 60°$ N at the equilibrium state from the linear analysis (a) and averaged between 10 and 100 days from the simulation start with the nonlinear solver (b).

simulation). The second jet is located at $\varphi_J = 60°$ N and has also a jet velocity of $U_J = 40 \ \mathrm{m\,s^{-1}}$ (the associated perturbation
field is shown in Fig. 10).

The perturbations resulting from the 30° N jet feature a combination of great-circle and along-jet propagation, that is properly represented by both the linear and nonlinear approaches (Fig. 9). On the other hand, for the jet at 60° N the along-jet propagation is obtained only in the linear simulation, while it is much weaker in the nonlinear simulation (Fig. 10). The weakening of along-jet propagation can be due to an enhanced equatorward propagation of the stationary wave in the nonlinear case, given that the
350 forcing is located at 45° N, possibly combined with enhanced dissipation. By using Eq. (20), the waveguidability of the 30° N jet is 84% while the one of the 60° N jet is 92%, highlighting that jets with same velocity act as similarly efficient waveguides.

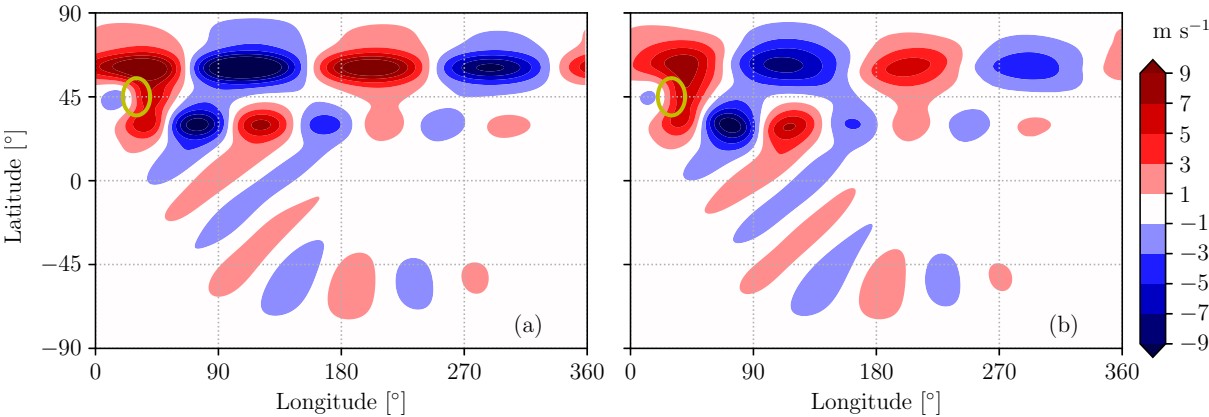

**Figure 11.** Meridional velocity associated with two zonal jets ($U_{J,1} = 40\,\mathrm{m\,s^{-1}}$ with $\varphi_{J,1} = 30°\,\mathrm{N}$ and $U_{J,2} = 40\,\mathrm{m\,s^{-1}}$ with $\varphi_{J,2} = 60°\,\mathrm{N}$) at the equilibrium state from the linear analysis (a) and averaged between 10 and 100 days from the simulation start with the nonlinear solver (b).

Moving to the double-jet configuration we notice, first of all, a good agreement between the linear and the nonlinear solution (Fig. 11). Rossby wave propagation occurs separately along the waveguides delineated by the two jet streams, while great-circle propagation to the Southern Hemisphere is smaller in the double-jet configuration than for the jets taken individually. The double-jet pattern corresponds roughly to the combination of the patterns of the linear solutions for the two individual jets. However, this is not true for the nonlinear solutions, because of the previously discussed lack of Rossby wave propagation along the 60° N jet in the nonlinear single-jet case. It might be that forced meridional velocity perturbations along the 30° N jet stream could provide energy to the jet at 60° N, which would otherwise be attenuated as in the single-jet case (Fig. 10b): however, this hypothesis would need further verification. Furthermore, the waveguidability of the two jets is lower than the one of the two distinct jets: the 30° N jet has a value of the waveguidability metric of 70%, while the 60° N jet has a value of 82%. The waveguidability of the 60° N jet has been estimated using Eq. (20), considering a fictitious forcing centered at 60° N and by monitoring the enstrophy anomaly around the same latitude (and similarly for the 30° N jet). The fact that the waveguidability is reduced compared to the isolated jets case by 10% for both jets is probably due to the jet streams' ability to attract leaked enstrophy from the surrounding regions (including the other jet stream), making the two jets more leaky than the ideal waveguides. However, the reader is warned that these results are likely sensitive to the choice of the reference state in Eq. (20) (here taken as the solid-body background flow).

## 6   Conclusions

Even though the basic physical concepts behind Rossby wave propagation have been known for several decades, we are still far from understanding Rossby wave evolution in non-idealized flow setups, such as the ones typically associated with extreme weather events. The comprehension of Rossby wave propagation along the waveguide provided by the upper-level jet stream,

in particular, remains a challenging task. This work revisited the problem of Rossby wave propagation in a nondivergent, barotropic flow on the sphere, following the specification of a zonally symmetric background flow. Such a simplified setup allowed to advance the state-of-the-art, both in terms of model and process understanding.

Firstly, we described a novel, linearised mathematical treatment of the barotropic Rossby wave propagation problem, that allows to efficiently obtain the steady-state Rossby wave response to a given topographic forcing. The approach relies on the framing of the barotropic vorticity equation on the sphere as an eigenvalue problem. Here, the eigenfunctions represent the flow response to the topographic forcing in the meridional direction only (as the background flow is zonally symmetric), while the sign of the imaginary part of the eigenvalues allows one to determine the stability of the eigenfunctions (a positive value indicates linear instability and exponential growth of the mode when triggered). In distinct contrast to ray-tracing theory, the present approach has general validity for any background flow and forcing combination, does not require projecting the flow field on a Mercator plane and does not assume any scale separation between the background flow and the waves.

Secondly, we elucidated some features of Rossby waves occurring in the barotropic model, in particular for what concerns the onset and the evolution of transient Rossby waves that, under some circumstances, accompany the steady-state response. Such waves are the result of a barotropic instability arising at high jet speed, when the complex part of the eigenvalue actually turns positive. The change of sign in the absolute vorticity gradient (i.e., the Rayleigh criterion) was not capable of detecting the onset of barotropic instability, which was observed at higher jet speeds than expected. The analysis neatly isolates the effect of dissipative conditions onto the onset of barotropic instability. Even more intriguing, nonlinear simulations showed signs of barotropic instability further beyond the linear neutral curve, pointing to potential damping effects operated by nonlinear terms (e.g., wave-wave interactions). This dynamics was further investigated by means of empirical orthogonal functions, which showed that modes with an azimuthal wavenumber of 5 up to 7 are associated with traveling waves of fixed amplitude leading to a limit cycle for generic initial conditions, a dynamics originated by nonlinear effects. Nevertheless, the averaged flow field in the nonlinear simulation resembles the unstable equilibrium state calculated from the linear method. The latter is an equilibrium state obtained by removing the time derivative and is therefore an approximation of the equilibrium condition of the nonlinear system, too.

Thirdly, we revisited the waveguidability problem, using the equilibrium state obtained from the linearized analysis to quantify the waveguidability for a variety of background flow configurations. A new metric of waveguidability, based on a time average of the enstrophy anomaly, has been proposed to compare jet streams located at different latitudes. We reconfirmed earlier results and noticed that, in a barotropic framework, jet waveguidability is strongly related to jet speed, while jet latitude plays a considerably smaller role. The application of the proposed approach to a double-jet configuration showed that: a) the equatorward propagation of Rossby waves is weakened in a double-jet case compared to the no-jet case; b) the double-jet response roughly corresponds to the linear combination of the individual jet responses although c) the waveguidability of each jet taken individually seems to be higher than when the two jets occur in a double-jet configuration. These results extend the the analysis by Wirth (2020), and pave the way for a more targeted analysis of Rossby wave propagation in presence of multiple jets, together with the development of appropriate metrics to quantify waveguidability in such circumstances.

We envisage several future applications of our analysis approach. For example, a systematic waveguidability assessment for different forcings and background zonal wind profiles, including a detailed investigation of double-jet configurations. Another relevant application would be to understand the role of orography in forcing Rossby waves with specific zonal wavenumbers, as amplified waves of specific wavenumbers have been related to surface weather extremes during summer (as noticed, among others, by Coumou et al., 2014; Jiménez-Esteve et al., 2022). Although we do not attempt this here, this evidence could enable building a reduced-order model of a barotropic atmosphere where the spatial modes are provided by the equilibrium state and by the most unstable modes (this information being retrieved exclusively from the linear analysis), while the temporal coefficients could be determined by solving a small set of nonlinear ordinary differential equations (Holmes et al., 2012).

In conclusion, we present a relatively simple and computationally efficient way to study the steady and unsteady Rossby wave response to topography in a variety of idealized background flow configurations. This approach goes beyond a number of limitations of ray-tracing theory for determining waveguidability, and we have illustrated the physical insights it can provide by considering a wide parameter sweep of jet speeds and latitudes and a double-jet configuration.

## Appendix A: Spectral solution of the barotropic vorticity equation

Equation (11) introduces the Laplace operator $\mathcal{L}$ in the transformed space together with the boundary conditions needed to avoid the pole singularity

$$\widehat{\psi}(\theta = 0) = 0 \quad \text{for} \quad m \neq 0 \quad \text{and} \quad \left.\frac{\partial \widehat{\psi}}{\partial \theta}\right|_{\theta=0} = 0 \quad \text{for} \quad m \neq \pm 1, \tag{A1}$$

and similarly for $\theta = \pi$. Consequently, there are two boundary conditions for $|m| \leq 1$ and four otherwise. An additional boundary condition must be introduced at $m = 0$ as $\widehat{\psi}(\theta = 0) = 0$ in order to set the value of the streamfunction at one pole since, otherwise, the streamfunction would be defined up to an additive constant, making the numerical problem singular.

Following Peyret (2002), the solution of Eq. (10) is calculated by a spectral collocation method in terms of orthogonal polynomials. In the present work, Chebyshev polynomials, $T_q(\theta)$, and the decomposition

$$\widehat{\psi}(\theta, t; m) \approx \sum_{q=0}^{N} a_q(t; m) T_q(\theta), \tag{A2}$$

are used. The $N+1$ collocation points are described by a shifted Gauss-Lobatto distribution

$$\theta_j = \frac{\pi}{2}\left[1 - \cos\left(\pi \frac{j}{N}\right)\right] \quad \text{with} \quad j \in \{0, 1, \ldots, N-1, N\}, \tag{A3}$$

implying a refinement of the colatitude distribution near the poles. This is in contrast with the Legendre and associated Legendre polynomials that account for a more appropriate resolution at the poles. Nevertheless, Chebyshev polynomials have been preferred here since an exact method to calculate the spectral coefficients exists for them, while the Legendre polynomials require a numerical quadrature scheme (Krishnamurti et al., 2006). The choice of the specific family of polynomials influences only the derivatives in the meridional direction and the meridional discretization, while the zonal discretization is still based on the Fourier transform as is the case for spherical harmonics.

Rather than working with the spectral coefficients, $a_q(t; m)$, here it is preferred to work directly with the value of the streamfunction at the collocation points, $\widehat{\psi}_j = \widehat{\psi}(\theta_j, t; m)$, as unknowns collected into a vector, significantly facilitating the interpretation of the results. The derivatives of the streamfunction are calculated as

$$\left.\frac{\partial \widehat{\psi}}{\partial \theta}\right|_j \approx D^{(1)} \widehat{\psi}_j \qquad , \qquad \left.\frac{\partial^2 \widehat{\psi}}{\partial \theta^2}\right|_j \approx D^{(2)} \widehat{\psi}_j \,, \tag{A4}$$

where the matrices $D^{(1)}$ and $D^{(2)}$ provide the first- and second-order derivatives (Peyret, 2002). Another advantage of the Chebyshev basis is that the matrices $D^{(1)}$ and $D^{(2)}$ are analytically known (Peyret, 2002; Canuto et al., 2006), although some modifications to limit the effect of round-off errors have been implemented (Bayliss et al., 1995).

By exploiting Eq. (A4), the discretized Laplace operator becomes

$$\mathcal{L} \approx D^{(2)} + \cot\theta \, D^{(1)} - \frac{m^2}{\sin^2\theta} I \,, \tag{A5}$$

where $I$ indicates the identity matrix. This implies that the Laplace operator becomes a numerical matrix since the spatial dependence of the variables is provided by the Fourier modes (in the zonal direction) and by the Chebyshev polynomials (in the meridional direction). Once the Laplace operator has been discretized, the matrices $A$ and $B$ in equation (12) are obtained from their definition.

As stated in Eq. (A1), the boundary conditions at the poles have been imposed by removing 2, 3 or 4 points from the analysis so that the size of the matrices $A$ or $B$ is $N - 2 \times N - 2$ for $m = 0$, $N - 1 \times N - 1$ for $|m| = 1$ and $N - 3 \times N - 3$ otherwise. This has the advantage that the reduced forms of $A$ and $B$ are just related to the dynamics of the modeled system rather than the boundary conditions (the reader is referred to section 3.7.1 of Peyret, 2002, for a detailed description on how the matrices can be reduced), together with a decrease of the computational cost.

If $B$ is nonsingular, it is possible to perform the eigendecomposition $B^{-1}A = P\Lambda P^{-1}$ where the columns of $P$ are the eigenvectors and $\Lambda$ is a diagonal eigenvalue matrix. The analytical solution of Eq. (12) is found to be

$$\widehat{\psi}_j(t; m) = Pe^{-i\Lambda t}\left[ P^{-1}\widehat{\psi}_j(0; m) + i\int_0^t e^{i\Lambda\tau} P^{-1} B^{-1} \widehat{F}(\tau) \, \mathrm{d}\tau \right], \tag{A6}$$

with $\widehat{\psi}_j(0; m)$ indicating the initial condition of the streamfunction. In the special case of a steady forcing, Eq. (A6) becomes

$$\widehat{\psi}_j(t; m) = Pe^{-i\Lambda t}P^{-1}\widehat{\psi}_j(0; m) + \left(I - Pe^{-i\Lambda t}P^{-1}\right) A^{-1}\widehat{F}. \tag{A7}$$

The first term of Eq. (A6) and (A7) is the homogeneous unforced solution of the linear system with initial condition $\widehat{\psi}_j(0; m)$, while the second term is the forced solution with zero initial condition.

Since the eigenvalues, $\Lambda$, and eigenfunctions, $P$, depend on the zonal velocity, $U$, Rossby number, Ro, and wavenumber, $m$, they are calculated and stored for all the considered wavenumbers for a given zonal velocity profile and Rossby number. Since they do not depend on the actual forcing applied, once they are computed they can be used with any kind of forcing according to Eqs. (A6-A7). Furthermore, since the equation is linear, no aliasing instability is present and each eigenmode does not interact with the others. Equation (A6) enables the calculation of the solution at any time without the need for numerical integration. This is nevertheless necessary if the forcing is time dependent, although with a definite integral instead.

## Appendix B: Evidence for a limit cycle behaviour in the unstable regime

In order to assess whether a limit cycle or a traveling wave takes place in the nonlinear simulation of the unstable strong-jet case, a series of targeted nonlinear numerical simulations have been performed. A long-time (200 days) simulation starting from the background state was performed and this is referred as the reference simulation in the following analysis. From the reference simulation it is possible to calculate the EOF modes of the streamfunction, namely the variable used to solve the barotropic vorticity equation. As in Fig. 5, the first mode $\Psi_0(\mathbf{x})$ is close to the time average of the forced system, while the successive modes $\Psi_1(\mathbf{x})$, $\Psi_2(\mathbf{x})$, ... are interpreted as unsteady Rossby waves. The projection of the instantaneous streamfunction field $\Psi(\mathbf{x}, t)$ onto the $n^{\text{th}}$ EOF mode provides the mode amplitude as

$$a_n = \int \Psi(\mathbf{x}, t) \Psi_n(\mathbf{x}) \mathrm{d}\mathbf{x}, \tag{B1}$$

where the integration is performed over the entire sphere surface.

The instantaneous values of the coefficients can be collected into a vector $(a_0, a_1, a_2, a_3, \dots)$ that provides a state space for the analysis. The EOF analysis of the reference simulation provided typical amplitudes of the EOF temporal coefficients $\sqrt{\langle a_{n,ref}^2 \rangle}$ and the EOF spatial basis, $\Psi_n(\mathbf{x})$, that is from now on kept constant in the following analyses.

Several nonlinear simulations were performed with different initial conditions given by

$$\Psi(\mathbf{x}, 0) = \sqrt{\langle a_{0,ref}^2 \rangle}\Psi_0 + \alpha_1 \sqrt{\langle a_{1,ref}^2 \rangle}\Psi_1 + \alpha_2 \sqrt{\langle a_{2,ref}^2 \rangle}\Psi_2 + \dots, \tag{B2}$$

namely by initiating our simulation from the time average field plus a set of Rossby waves with arbitrary amplitudes. If the state space evolution from different initial conditions will lead to the same orbit, a limit cycle is present, while if the orbit depends on the initial condition a traveling wave is present. Figure B1 shows the evolution of the $a_1 - a_2$ and $a_3 - a_4$ coefficients for four different initial conditions, that nevertheless lead to orbits that spiral towards the same region, supporting the presence of a limit cycle.

It appears that if the modes have a too high amplitude, they will be damped, while for too small amplitude they will amplify as predicted by linear theory. This is visible for the $a_1 - a_2$ coefficients, while the $a_3 - a_4$ coefficients undergo a transient growth first and decay afterward.

*Author contributions.* AS developed the theory, performed the simulations shown in the manuscript and analyzed the data. All authors contributed equally to the interpretation of the results. AS and JR prepared the manuscript, assisted by GM and VW for review and editing. The authors are thankful to the two anonymous reviewers for very insightful comments during the revision process.

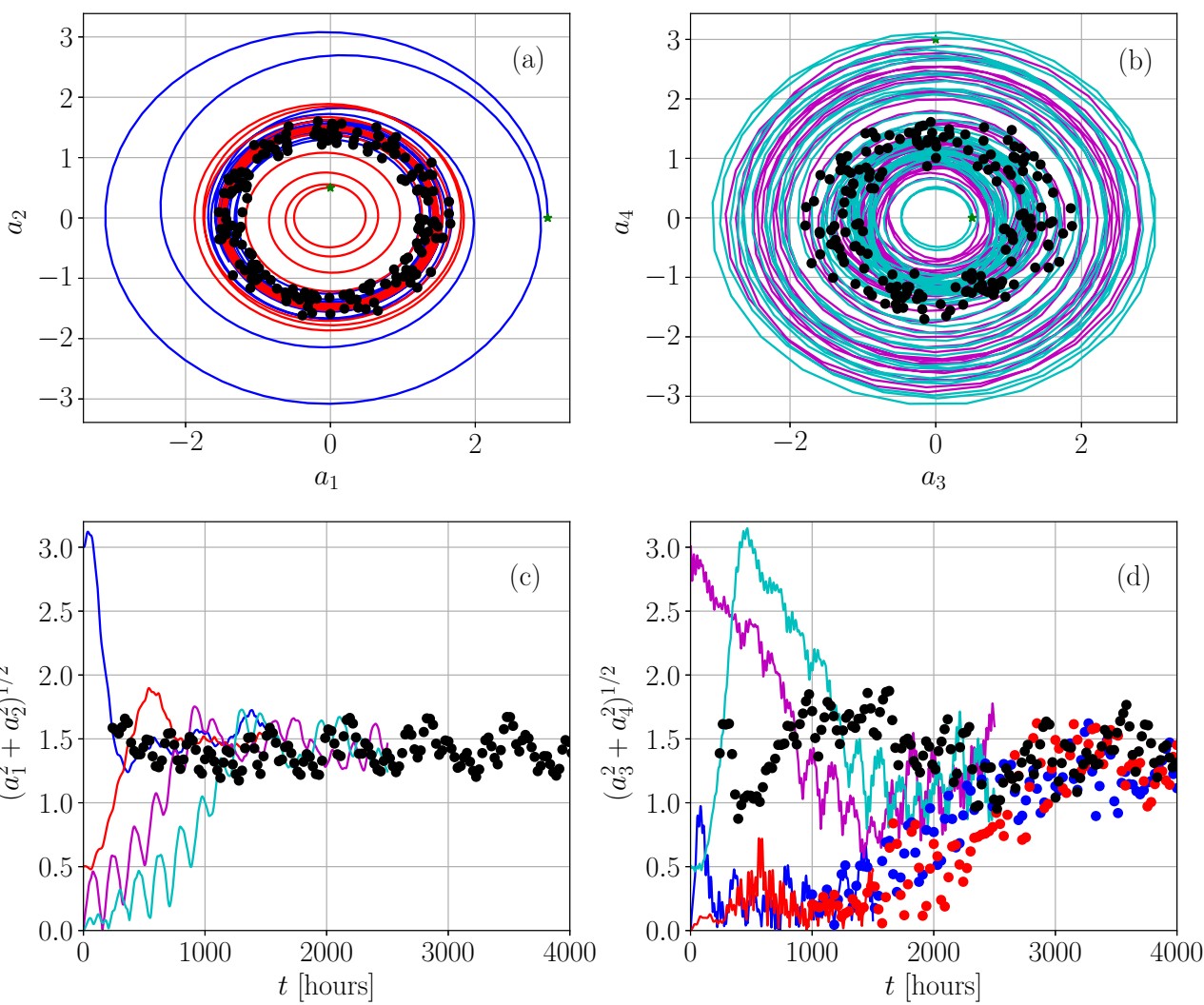

**Figure B1.** State space sections. (a) $a_1 - a_2$ coefficients, (b) $a_3 - a_4$ coefficients. (c) $\left(a_1^2 + a_2^2\right)^{1/2}$ temporal evolution, (d) $\left(a_3^2 + a_4^2\right)^{1/2}$ temporal evolution. Black circles: state space evolution over 4000 hours with initial condition $\Psi(\mathbf{x}, t = 0) = 0$. Blue: initial condition according to Eq. (B2) for $\alpha_1 = 3$. Red: $\alpha_2 = 0.5$. Magenta: $\alpha_3 = 3$. Cyan: $\alpha_3 = 0.5$. The starting point of the lines is highlighted by a green asterisk. Solid lines correspond to short simulations with high sampling frequency, while the circles are longer simulations with lower sampling frequency.

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
