# Peer review of "A linear assessment of barotropic Rossby wave propagation in different background flow configurations"

_EGUsphere, 2023_

## Referee Comment (RC2)

Review of egusphere-2023-316

"A linear assessment of waveguidability for barotropic Rossby waves in different large-scale flow configurations"
by
Antonio Segalini, Jacopo Riboldi, Volkmar Wirth, and Gabriele Messori

**Recommendation: Reject - consider resubmission to a more technical journal after major revisions**

**General Comments:**

While the manuscript is well written and the figures well prepared, it is not fully clear what this paper is about. The title and introduction suggest that the manuscript is about an assessment of waveguidability for different background flows, whereas the results mainly focus on introducing a solution technique for Rossby waves, which is not entirely novel in its design. The discussion and conclusions leave the reader wondering how the introduced methodology is aiding the overall question on waveguidability for different flow configurations, as only very few highly idealized setups are tested. Hence, given the more technical character of the manuscript and lack of presentation of direct scientific usage of the method, this manuscript is not suited for Weather and Climate Dynamics in its current form and a resubmission to a more technical journal, such as Geoscientific Model Development, should be considered after major revisions have been implemented.

The very notion of waveguidability defined as in line 31 demands an a priori philosophical choice about the separability of the atmospheric state into a "basic state" and "wave perturbations". In the beginning of the second paragraph, the authors first emphasize the relevance of waveguidability for extreme weather events but at the end of the same paragraph the authors state that the assumption of separating the perturbation from the basic state is violated during extreme events. What should the reader take home from this obvious contradiction? Are there particular limits that the authors would like to point the reader to?

In the final paragraph of the introduction, the authors state the actual content of the paper, though the very aspects that they test are not really motivated in the previous paragraphs of the introduction. What is the actual question at hand? What is the context of this study? If this study is only about the linear solution for various basic states, one won't be able to address the conundrum pointed out in the second paragraph, and in a way most of the wave refraction arguments have already been put forward in previous publications two to three decades ago. The comparison to the non-linear simulation can provide an assessment to the limits of the analytic solution, though it is not assessed in greater detail in the manuscript vis-à-vis the limitations of philosophical choices to thinking in a framework of waveguidability.

From line 109 onwards the authors state that the question arises as to how the equilibrium state is obtained and subsequently mainly address the homogenous time-evolving part of the solution that does not project onto the forcing and thereby the equilibrium state. This is rather confusing, as these transient modes, stable or not, will not project onto the

stationary forcing and the equilibrium state. It is thus unclear what the authors try to achieve and construct. In the ensuing section, indeed only the forced response is focused on again. The authors need to more clearly outline the rationale of their work, as it is currently difficult to follow what they aim to achieve.

Also in the results sections the focus is more on the actual method and its performance when compared to other solution techniques. The authors also include a discussion on the influence of model resolution on the performance of their method. While this is all interesting and relevant, it again emphasizes the more technical character of this manuscript with an absent focus on actual applications to more general background flows.

For the strong single jet case, the authors discuss unstable solutions and even perform an EOF analysis on the non-linear simulation. The relevance of this to the presented solution technique is unclear. The authors discuss some of the linear unstable modes in the light of the identified EOFs, though indicate that the matching is not convincing. The ensuing subsection on stability analysis is therefore also difficult to contextualize with the rest of the manuscript. In particular, it is unclear if the authors present the unstable modes to discuss instability, or if they present the unstable modes to assess the validity of their linear method. This confusion relates back to the general comment further above about the general topic of the manuscript being unclear.

The double jet discussion is interesting and in fact one of the parts of the manuscript that also makes a scientific contribution beyond the technical aspects. However, most of the findings there are not necessarily new or unexpected and should thus be put in context to existing literature on wave refraction, ducting, and tunneling.
Overall, it is not clear how the presented approach is novel or how it yields additional information compared to more traditional linear approaches to assess wave propagation, such as the method the authors compare their results to (spectral harmonical method). If their method is arguable superior to existing methods, this should be made clearer in the manuscript.

**Specific Comments:** (Reference to line numbers in the manuscript)

L15: Hoskins and Ambrizzi (1993) should be stated, as it is probably the most classical reference in this context.

L16: Wave guiding also goes back to the early work on refraction of Rossby waves, so this sentence reads a bit redundant in the light of the previous sentence.

L10-29: The first paragraph is rather long and the main topic is not clear. The paragraph might benefit from splitting it and more clearly addressing the context for this manuscript.

L31: Waveguidability is explained here for the first time, while the reader is left wondering during the first paragraph about its meaning.

L61: Do the authors really mean "stability" in the sense of a wave instability or in the sense of applicability of the linear analytic solutions?

L74: It is confusing to refer to Lambda as longitude, which is not even used in the equations thus far, while at the same time using Lambda_r as a dampening parameter. The authors are encouraged to change the naming of the dampening parameter to avoid confusion.

L147-153: Almost everything stated here is not new, even though the authors make it sound like a new discovery. Previous findings should be clearly stated and referenced.

L163-173: It is not made clear to the reader why this resolution sensitivity study is performed and its relevance to the assessment of waveguidability.

---

## Author Comment (AC1)

A linear assessment of barotropic Rossby wave propagation in different
large-scale flow configurations
by
A. Segalini, J. Riboldi, V. Wirth & G. Messori

Comments to Reviewer #1:
*(the text of the reviewer is in italic)*

We appreciate the feedback regarding our manuscript. In the following
we address the reviewer's suggestions for improvement, and point out the
changes compared to the original manuscript. Parts that have been rewritten
or added due to comments by the referees have been highlighted in red in
the revised version of the manuscript.

> *I think the method and results presented in this manuscript are
> interesting and significant, and could be useful for future studies.*

We thank the referee for the support to our work.

> *I personally had to read through the manuscript carefully twice
> before I had a sense of what it is about.*

We have extended the introduction and extensively modified Sect. 4–7
to better motivate the work (both in terms of introducing the methodology
and of highlighting which new scientific insights result from our analysis) as
both referees pointed out that the previous manuscript had a structure that
hindered readability.

> *1) In my opinion, the introduction, the abstract and perhaps even
> the title do not express clearly what the paper is about. They
> give the impression that the paper is more about the physics of
> waveguides, whereas the main contribution of the paper, as I see
> it, is the computational method. The single and double jet cases
> shown in the paper are used as test cases for the computational
> method, rather than going deep into the dynamics of waveguides.*

We agree with the referee that part of the novelty of the study lies in
the new computational approach. However, we would also like to highlight
that the implementation of such approach provides novel physical insights

(see e.g. the analysis of the double jet in Sect. 5). We thus believe that the contents of the study cannot be fully appreciated without an understanding of the current state of research on Rossby waveguides. The intertwining between methodology, theory and physical insights is particularly relevant for this topic, as a universally accepted definition of waveguide is still lacking. White et al. (2022) and Wirth and Polster (2021) showed that our capability to "see" waveguides critically depends on the methodological approach chosen to diagnose them. We perhaps described this problem in excessive detail in the Introduction, with the effect of de-emphasizing the methodological novelty of our approach. We have now added a new paragraph to the introduction focusing on methodologies for waveguide analysis in the literature. In parallel with this, we have extensively modified the results and concluding sections (including a new Sect. 6) to better highlight the physical insights gained from the proposed linear analysis. Finally, we have revised the title to better reflect the focus of the paper, which is on wave propagation rather than waveguidability.

> *In the last paragraph of the paper the authors write that "This study should be regarded as an introduction and explanation of the techniques, but possible applications of this approach could include systematic waveguidability assessments for different forcings and background zonal wind profiles." I think this sentence should appear in the introduction, and the abstract should emphasize the main point of the paper.*

The paper has been partially re-written to better address the final outcomes of the paper. In particular:

- The waveguidability has been discussed more by means of the definition proposed by Wirth (2020);

- The improvements compared to ray-tracing theory (limited by slowly-varying background flow, steady wave propagation, Mercator projection) have been described in the introduction, in section 2 and in the manuscript concluding section;

- The difference between the classic barotropic stability criterion, linear neutral curve and locus where the nonlinear simulations become unstable has been extended in section 4.2.

*2) Perhaps the authors could add some more motivation for specific choices in the derivation of the mathematical model, that could add to the clarity of the paper. One choice that wasn't immediately clear to me was including a damping term in equation (1). It wasn't clear what this damping term represents physically. It was also not clear to me what the motivation was for looking at the stationary solution in equation (13). Only after I finished reading it became clearer. I think that in section 2 it could help to explain better what the model is supposed to represent, before the derivation of the equations.*

We thank the reviewer for this suggestion. It is quite common to introduce this kind of damping term (see for instance Hoskins and Karoly, 1981, Hoskins and Ambrizzi, 1993, Wirth, 2020). Yet, we agree that its use should be explained and that we did not elaborate this important term adequately in the previous version of the manuscript. The latter has been updated according to the suggestions of the referee around equation (1) and (13) to clarify the meaning of the attenuation term and of the equilibrium solution. Furthermore, following the suggestion of the other referee, we have changed the symbol of the attenuation parameter from $\lambda_r$ to $\chi$.

*3) The authors present stationary solutions and time-dependent solutions, but it is not mentioned explicitly at which section each type of solution is examined, what each type of solution represents physically, what is the motivation for looking at each type of solution and what are the different methods for solving for a stationary or time-dependent solution. Each of these are explained somewhere, often after presenting the results, but it is not explained in an organized manner.*

Our aim in the results section was to highlight that the linearised methodology provides a simple and reasonably good assessment of the temporal evolution of the waves, while the majority of the analyses in the literature were focused on the equilibrium solution (for the linearised system) or on the temporally averaged nonlinear numerical solution. We nonetheless understand the confusion our previous framing of the results caused. We have now rearranged the structure of the results to separate the discussions of the stationary solution from the time-dependent solutions. In particular, we

have moved the discussion of the unsteady case at the end of section 5 as a new section about time-dependent analyses. While the procedure to obtain the stationary solution is described in Sec. 2 [Eq. 12], the derivation of the temporal analytical solution (in the sense that no temporal discretisation is needed) is provided in the paper appendix with both steady and unsteady forcing.

> *4) Section 4.3 analyzes the stability of the problem for different parameters of the jet profile. I was missing a discussion on the connection of this instability to linear barotropic instability, in the sense of the necessary conditions for instability including a change of sign of the PV gradient. Some more physical context would be useful.*

We thank the referee for this comment. By repeating the derivation of the Rayleigh stability criterion it is found that a necessary condition for the instability is that the PV gradient becomes zero for the spherical case as well. However, this condition is not sufficient, since it enables the imaginary part of $\omega$ to be different from $-\lambda_r$ ($-\chi$ in our current notation), so that there is still a large stability margin until the imaginary part of $\omega$ becomes positive and the mode unstable. This is actually what we observe from the present analysis since the PV gradient changes sign when $U_J = 9$ m/s (for a jet at 45° N and $\sigma_J = 5°$), while the flow becomes linearly unstable at around $U_J \approx 20$ m/s. In the stability analysis part we have now discussed this and updated the figure with the eigenvalue and the neutral curve to show the curve where the PV gradient changes sign: as visible in the new figure 8 (figure 1 of the present reply), the Rayleigh criterion is violated in a region that is still stable. If the damping was absent, then the Rayleigh criterion will become a sufficient condition for the instability. We have added this discussion in both Sect. 4.2 and Sect. 7 of the revised version of the paper.

> *5) The analysis of the double jet case shown is much shorter than that of the single jet case and does not include a calculation of waveguidability. Based on the last sentence of the abstract ("Examples using single- and double-jet configurations are discussed to illustrate the method and study how the background flow can act as a waveguide for Rossby waves") I was expecting a comparison between the waveguidability of the two cases. If the authors choose*

[Figure]

Figure 1: (a) Maximum of the imaginary part of the eigenvalues for a given jet velocity (normalised by $\chi$) when $\sigma_J = 5°$. The black line indicates the neutral curve, the dashed line is the locus where the absolute vorticity gradient changes sign (Rayleigh stability criterion), while the green thick line is the locus beyond which the temporal vorticity variance becomes 10 times larger than in the stable regimes (approximating the neutral curve in the nonlinear case). (b) Azimuthal wavenumber of the most unstable eigenvalue for $U_J = 40$ m/s for different jet widths. The black lines are curves $m \propto \cos \varphi_J$ fitting the linear stability results.

> *not to include a calculation of waveguidability for the double jet case, they should otherwise motivate the choice for the analysis that is presented.*

We have followed the suggestion of the reviewer and have now assessed the waveguidability of both jets in the revised version of the manuscript, always using the diagnostic by Wirth (2020). These results have been added to Sect. 5 as a new figure 3 (figure 2 of the present reply). The waveguidability of the two jets taken separately is around 37% for the southern jet and 58% for the northern one, so that the northern jet has more waveguidability than the southern one. When two jets are simultaneously present, the northern jet shows again an even higher waveguidability (98%) compared to the southern jet (71%). This is not necessarily an intuitive result and highlights the type of physical insight that our approach can provide.

> *The term "analytical solution" used in the abstract and introduction was confusing for me. I expected to see a solution expressed*

[Figure]

Figure 2: Waveguidability of the single jet zonal profiles for different jet latitudes and strength. Only narrow jets with $\sigma_J = 5°$ have been considered.

> *as a mathematical function. It is true that the Chebyshev polynomials are analytical functions and in that sense the solution is analytical, but eventually there is a numerical calculation that leads to the solution, so perhaps a different terminology would describe the method more clearly. As I see it, the main difference between what is called a "numerical solution" and "analytical solution" by the authors is that the former is a time-integrated solution and the latter is an eigenvalue problem.*

We agree and we have removed the adjective "analytic" from the majority of the manuscript where a numerical assessment was involved. As the referee acknowledges, Chebyshev polynomials are just a basis for the projection of the streamfunction and they are characterised by an exponential convergence as typical of spectral methods. However, the analytical form of the solution (obtained with pen and paper) is absent even if the solution is nearly exact.

> *Lines 181-182: In what sense is it counterintuitive that the nonlinear solution is more dissipative?*

The presence of nonlinearities is associated with turbulent effects by the onset of an energy transfer towards other scales. This energy transfer across scales often leads to the divergence of the solution and to the onset of turbulence in fluid dynamics problems (e.g., flow in a tube, Kundu & Cohen, Fluid

Mechanics, 2013), but instead here the nonlinear terms appear to smooth the solution. This is also pointed out by the stability analysis. In the linear case with narrow jet ($\sigma_J = 5°$ and jet latitude at 45° N) the flow becomes linearly unstable with a jet intensity of $U_J \approx 20$ m/s, while the nonlinear simulation becomes unstable at $U_J \approx 28$ m/s, highlighting an unexpected stabilizing effect of the nonlinear terms. We have removed the adjective counterintuitive in the revised version of the manuscript since the damping/amplifying effect of nonlinearities is probably an heuristic feature that requires more investigation.

> *Perhaps the appendix can include some more details of the computational method, such as the matrices D(1) and D(2), and how the time-dependent solution is calculated.*

The matrices $D_1$ and $D_2$ are obtained by considering the first-order derivative of the Chebyshev polynomials (grouped as a matrix $T$ where the columns are the polynomials and the rows are the colatitude locations). The derivative polynomials can be collected by means of a matrix $T_1$. The matrix $D_1$ can be written as $D_1 = T_1 \cdot T^{-1}$ so that the matrix $T^{-1}$ converts the streamfunction distribution into the associated polynomial coefficients and $T_1$ provides the derivative in physical space. This discussion is not strictly necessary in the article and we fear that including it would shift the focus on the numerical details rather than on the methodology, so we prefer to omit that. However, we provide a reference to Peyret that goes sufficiently in detail about estimates of these matrices and on how round-off errors can be reduced.

Regarding the time integration, we provide details in equation (A7): The matrices $A$, $\Lambda$ and $P$ (together with their inverses) are computed in the pre-processing stage once and for all, together with the Rossby waves features. At any arbitrary time $t$ only a few matrix multiplications are needed to obtain the solution evolution in the stable case (in the unstable case it works well too until the unstable eigenmodes become too large). No time integration (or time stepping) is needed in the linear stable case, and we now explicitly state this in the Appendix.

---

## Author Comment (AC2)

A linear assessment of barotropic Rossby wave propagation in different
large-scale flow configurations
by
A. Segalini, J. Riboldi, V. Wirth & G. Messori

Comments to Reviewer #2:
*(the text of the reviewer is in italic)*

We appreciate the feedback regarding our manuscript. In the following
we address the reviewer's suggestions for improvement, and point out the
changes compared to the original manuscript. Parts that have been rewritten
or added due to comments by the referees have been highlighted in red in
the revised version of the manuscript.

> *While the manuscript is well written and the figures well pre-*
> *pared, it is not fully clear what this paper is about. The title and*
> *introduction suggest that the manuscript is about an assessment*
> *of waveguidability for different background flows, whereas the re-*
> *sults mainly focus on introducing a solution technique for Rossby*
> *waves, which is not entirely novel in its design. The discussion*
> *and conclusions leave the reader wondering how the introduced*
> *methodology is aiding the overall question on waveguidability for*
> *different flow configurations, as only very few highly idealized se-*
> *tups are tested. Hence, given the more technical character of the*
> *manuscript and lack of presentation of direct scientific usage of*
> *the method, this manuscript is not suited for Weather and Climate*
> *Dynamics in its current form and a resubmission to a more tech-*
> *nical journal, such as Geoscientific Model Development, should*
> *be considered after major revisions have been implemented.*

Although we agree with the reviewer's critique that the structure and the
aims of the paper can be made more intelligible, we believe that the novelty
of our contribution is not exclusively methodological (see for example the
double jet results presented in Sect. 5). Indeed, what we seek to do is
propose a new paradigm to study the flow evolution of forced Rossby waves
that rapidly enables to investigate in principle any background flow and
forcing combination. As far as we are aware, the "very few idealised setups"
we present are currently the largest systematic collection of background flows
investigated in any single paper in the literature (see e.g. the new Fig. 3

[Figure]

Figure 1: Waveguidability of the single jet zonal profiles for different jet latitudes and strength. Only narrow jets with $\sigma_J = 5°$ have been considered.

of the manuscript, namely Fig 1 of the present reply). Moreover, since our study specifically speaks to the meteorological community, we believe that Weather and Climate Dynamics is the best-suited outlet for its publication. The consideration of idealised setups should not be an issue per se, as WCD's scope explicitly includes "idealized numerical studies", by the journal's own description.

To better understand the innovative character of our approach, we compare it to the classic ray-tracing approach proposed 40 years ago by Hoskins and Karoly (1981), and still used in the literature today. The ray-tracing technique requires the tracking of the wave during its evolution. However, a wave written in the form

$$\psi = \widehat{\psi} \exp\left[i\left(kx + ly - \omega t\right)\right] , \tag{1}$$

is correct only when the background flow $U$ is constant in the $\beta$ plane. This requires several approximations to apply this simple theory to general flow over a sphere:

1. A Mercator projection of the flow field

2. The background flow must be slowly varying compared to the spatial scale given by the wavelength of the Rossby wave

3. Although not an intrinsic limitation, in practice the analysis is typically conducted on stationary waves

Under these assumptions we can use WKB theory and extend the classical Rossby solution to more realistic cases with jet streams, for instance. However, we lose information about how the waves are evolving in time or whether the flow will ever approach a steady state: Wirth (2020) noted that, when a strong jet stream was present, the Hovmöller diagram did not approach a steady state and filtering was required to get it.

The approach that we propose is completely different from previous works since any generic zonal background flow can be considered and no ray tracing is required. The waves are obtained as a result of the equations independently from the forcing so that general conclusions can be obtained. No assumption of scale-separation is required nor do we need to project the flow in a Mercator plane, with the consequent deformation at the poles. Furthermore, a stability analysis can now be performed systematically, enabling an estimate of the flow evolution. Even the interpretation of the nonlinear simulation results is much facilitated by the proposed linear framework, for example in terms of obtaining a lower bound for (nonlinear) instability onset. Therefore, we see the added value of this work for the atmospheric dynamics community as a paradigm shift in wave propagation analysis. While the technical details of our approach are essential to ensure reproducibility of the study, they should be viewed as functional to the proposed new wave analysis paradigm, and not as a key result in themselves. To address the Reviewer's concerns, we have now reworded the final two paragraphs of the introduction to frame more clearly the motivation for our work. We have further modified the title to shift the focus from waveguidability to wave propagation, which is the core of our results. The new title now reads: "A linear assessment of barotropic Rossby wave propagation in different large-scale flow configurations". Finally, we have clarified earlier on in the introduction the key limitations of ray-tracing that our approach overcomes.

> *The very notion of waveguidability defined as in line 31 demands an a priori philosophical choice about the separability of the atmospheric state into a "basic state" and "wave perturbations". In the beginning of the second paragraph, the authors first emphasize the relevance of waveguidability for extreme weather events but at the end of the same paragraph the authors state that the assumption of separating the perturbation from the basic state is violated*

*during extreme events. What should the reader take home from this obvious contradiction? Are there particular limits that the authors would like to point the reader to?*

As the reviewer notices, we are indeed trying to point the reader to a contradiction in how previous research on the topic has been conducted. The first paragraph introduces Rossby waveguides and discusses how they have been invoked by several studies as decisive contributors to recent extreme weather events. The second paragraph, on the other hand, cites research work that has pointed out the limitations of available diagnostics and theoretical frameworks to diagnose Rossby waveguides. For instance, Wirth and Polster (2021) noticed how the methods used in the previous studies can lead to misleading conclusions, pinpointing to the presence of waveguides even if there aren't any. This critical review of the literature aims at highlighting this contradiction to the reader and at justifying attempts to move forward, towards an improved understanding and a shared definition of Rossby waveguides.

We agree with the reviewer that any wave analysis requires the separation of the flow into a background component (usually steady or slowly changing in time and in space) and a wave component (as a perturbation that is faster in time and characterized by smaller spatial scales). However, the question about how we can identify the background flow remains unsolved and it is not our aim to address this very important task. Wirth and Polster (2021) have recently investigated this problem by applying various spatial and temporal filters to artificially perturbed potential vorticity fields to determine the ability of these methods to identify the (known) background flow. It was noted by Wirth and Polster that standard average methods work well when disturbances have small amplitudes and are not persistent, while large-amplitude waves influence very much the identified background flow. If the background flow is erroneous, the associated waves (obtained from the present analysis for instance) are erroneous too, underlining the high importance of this task. Once again, it is not our aim to identify the actual background flow from a snapshot of ERA5 data, for instance. Instead, the goal of this work is to assess the wave properties given the background flow where they operate. In this respect, our paragraph in the introduction is informative and provides a note of caution about a task that should not be underestimated.

*In the final paragraph of the introduction, the authors state the actual content of the paper, though the very aspects that they test*

*are not really motivated in the previous paragraphs of the intro-duction. What is the actual question at hand? What is the context of this study? If this study is only about the linear solution for various basic states, one won't be able to address the conundrum pointed out in the second paragraph, and in a way most of the wave refraction arguments have already been put forward in pre-vious publications two to three decades ago. The comparison to the non-linear simulation can provide an assessment to the limits of the analytic solution, though it is not assessed in greater detail in the manuscript vis-à-vis the limitations of philosophical choices to thinking in a framework of waveguidability.*

As outlined in our first reply, the approaches currently used in the lit-erature for the study of waveguidability have a number of conceptual and practical limitations (we highlight in particular three main ones on p. 2 of this reply document). One way to phrase the scientific question behind our work could be: is it possible to overcome these limitations? In this study, we set out to develop a new paradigm for the study of flow evolution of forced Rossby waves, which provides a computationally efficient, forcing-independent wave solution applicable in principle to any background flow. Crucially, this also overcomes the above-mentioned limitations present in the literature. The benefits of such an approach compared to the state of the art on the topic are multiple, as again outlined in our first reply. Concerning the link to the introduction, we believe that an introduction should provide context and motivation for the work, beyond a simple list of points that will be addressed in the analysis. We have nonetheless shortened the intro-duction and limited the discussion of background flow derivation, to provide a more focused text in the spirit of the Reviewer's comment. Concerning the Reviewer's comment on the linear solution, we argue that even without invoking the comparison to the nonlinear case, our results provide new un-derstanding compared to the literature. Specifically, we can understand how one approaches the steady-state solution and even infer physical information about the temporal evolution of the waves when there is no steady state (e.g. in the strong jet case). To our knowledge this is not something presented in previous analyses. The linear simulation additionally facilitates the inter-pretation of the nonlinear case by providing a lower bound for the onset of (nonlinear) instability. We see the above, and in particular the inferences we make on the nonlinear (in)stability from the linear analysis, as independent

of a waveguidability assessment. Indeed, the very concept of waveguidability is complementary to having a full wave solution.

> *From line 109 onwards the authors state that the question arises as to how the equilibrium state is obtained and subsequently mainly address the homogenous time-evolving part of the solution that does not project onto the forcing and thereby the equilibrium state. This is rather confusing, as these transient modes, stable or not, will not project onto the stationary forcing and the equilibrium state. It is thus unclear what the authors try to achieve and construct. In the ensuing section, indeed only the forced response is focused on again. The authors need to more clearly outline the rationale of their work, as it is currently difficult to follow what they aim to achieve.*

We agree with the reviewer's comment and we modified the structure of the results to address it. In particular, we now discuss separately the equilibrium state and the transients (now shifted to section 6), avoiding to jump back and forth between the two. Hoping to further clarify the results, we discuss now why the stability analysis and the analysis of the transients are important elements in the rationale of our work. The equilibrium point can be conveniently obtained by eliminating the time derivative and therefore looking only at the stationary solution. This was already done by Hoskins and Karoly (1981) for instance. However, if the steady-state solution is linearly unstable, then this state will not be achieved (within the linear formulation). Therefore, inspired by dynamical system theory, we named this solution an equilibrium state where the temporal derivative is zero. If this state is stable, the system will evolve to approach the equilibrium solution regardless of the initial condition (for a linear system there is only one equilibrium state). Otherwise, the linear system should diverge from that. Our interest is not the determination of this state but rather if it is ever possible to achieve and how. One of the most surprising discoveries in this work has been the observation that, even if the equilibrium state is unstable, the state of the nonlinear system oscillates around this equilibrium condition.

The comment of the referee about the homogeneous/forced part is partly correct. It is true that the modal basis is obtained for a homogeneous system, but it is also true that the forced system response is obtained by means of the variation of constants method where the same functions are exploited to get

the solution (A6). Equation (A7) provides the full solution of the steadily-forced system. When looking at the time-dependent forced response, this is once again known analytically for the linear system once the eigenfunction and eigenvectors are known. We intended to show this by including the new figure 12 (figure 3 in the previous version of the manuscript) where the temporal evolution of the solution is shown. While most of the present work is still focused on the equilibrium state, we felt that it was important to highlight that our approach also enables a study of the temporal evolution.

> *Also in the results sections the focus is more on the actual method and its performance when compared to other solution techniques. The authors also include a discussion on the influence of model resolution on the performance of their method. While this is all interesting and relevant, it again emphasizes the more technical character of this manuscript with an absent focus on actual applications to more general background flows.*

We hope that the modifications we are bringing to the revised manuscript will further emphasize the physical aspects behind Rossby waveguides, rather than the technical details. The grid-convergence analysis as well as the comparison with the solid-body velocity distribution $U_\lambda = 15\sin\theta$ (where the analytic solution of the linear problem is known) is placed at the beginning of section 3 "Model validation" to validate the model, namely to assess its consistency with what known already from the literature. The comparison of the waveguidability estimated by Wirth (2020) by means of nonlinear simulations is a warning to the reader to keep in mind that linear and nonlinear estimations are not always the same and, in particular, to highlight that linear simulations provide qualitative trends that are easy to interpret in light of the system's linearity.

We have tried to limit the technical details within the manuscript as much as we could,moving some more technical parts in the paper appendix. However, we are keen to ensure full reproducibility of our results, and thus believe that it is important to have a detailed description of the methodology, even if this may partly dilute the focus from the physical insights obtained by its implementation. Once the methodology is introduced, it provides a new way to interpret linear and nonlinear simulations and to understand the wave evolution in a more general and systematic framework that we believe to be valuable.

We recognize that we did not discuss sufficiently waveguidability in a general context and we have now included the new figure 3 in the revised manuscript (also provided as figure 1 of the present reply document) that provides the waveguidability for a variety of single jet latitudes and strengths. We highlight that such a figure would have been very demanding to produce without the new approach we propose. Furthermore, we have now reported the waveguidability values in the double jet case in Sect. 5. We hope that these improvements highlight the physical insights brought by our analysis, and the fact that our study goes beyond the simple description of a new method.

> *For the strong single jet case, the authors discuss unstable solutions and even perform an EOF analysis on the non-linear simulation. The relevance of this to the presented solution technique is unclear. The authors discuss some of the linear unstable modes in the light of the identified EOFs, though indicate that the matching is not convincing. The ensuing subsection on stability analysis is therefore also difficult to contextualize with the rest of the manuscript. In particular, it is unclear if the authors present the unstable modes to discuss instability, or if they present the unstable modes to assess the validity of their linear method. This confusion relates back to the general comment further above about the general topic of the manuscript being unclear.*

The EOF analysis is presented to give information about the temporal evolution of the flow. Wirth (2020) noted that the strong jet case exhibited oscillations that did not decay even after a long time. He solved this point by taking the temporal average of the simulation, as we also did. The comparison between the temporal average and the equilibrium state was judged reasonably good and is reported in the manuscript. At the same time, it was strange to admit that, according to our analysis, the strong jet case was linearly unstable and therefore it should diverge from the equilibrium state, which it did not in the nonlinear case (what we consider as the ground truth). In order to explain this paradox, we started looking at the EOFs as a way to simplify the temporal variation now summarized by the modal coefficients. The first mode was provided by the time-averaged solution, while all the other modes were traveling waves centered at the jet location (irrespective of the forcing location), and their topology is strikingly similar to the most unstable modes. A discussion about this is now reported in the revised version of the manuscript in Sect. 4.1.

The characteristic of these traveling waves is actually not relevant if only the time-average state is of interest (as in ray-tracing theory). However, at any instant the unstable linear modes should be the ones growing the most and receiving the highest energy from the background flow. In a linear system they should just grow unbounded, while in a nonlinear system their growth should not be unbounded and their energy is transferred to other wave components by nonlinear interactions. Most likely we were expecting that the linear unstable modes will keep competing to receive the transferred energy and start growing again. Therefore, we wanted to point out that the linear analysis, much less useful in the unstable regime, was still providing useful insight since the first EOF modes (the first is just the mean in our analysis) were actually very similar to the most unstable waves identified in the linear analysis. Practically, this implies that the most energetic EOF modes are approximately known from the linear analysis and one does not need to compute them a posteriori with the EOF methodology, enabling the development of a reduced-order model. We now mention this possibility at the end of Sect. 4.1.

> *The double jet discussion is interesting and in fact one of the parts of the manuscript that also makes a scientific contribution beyond the technical aspects. However, most of the findings there are not necessarily new or unexpected and should thus be put in context to existing literature on wave refraction, ducting, and tunneling.*

Following the referee's suggestion, we decided to extend this section of the result: we calculated waveguidability also for double-jet configurations and included a new figure for the single-jet case (new Fig. 3). As far as we are aware, neither of these results has previously been presented in the literature. For example, despite the large interest in the subject (e.g., Rousi et al. 2022), to our knowledge there hasn't been any quantitative assessment of waveguidability for double jet configurations. The resulting waveguidability values are, at least for the authors of this manuscript, not necessarily intuitive. We thus believe that the results in the revised manuscript are indeed new and partly unexpected.

> *Overall, it is not clear how the presented approach is novel or how it yields additional information compared to more traditional lin-*

*ear approaches to assess wave propagation, such as the method the authors compare their results to (spectral harmonical method). If their method is arguable superior to existing methods, this should be made clearer in the manuscript.*

We have already mentioned that the proposed methodology goes beyond limitations of the ray-tracing theory, namely that

1. steady and unsteady waves can be discussed

2. waves without scale separation from the background flow can be analyzed

3. no need to perform Mercator projections without the consequent deformation at the poles

We feel indeed that new physics can be investigated with more confidence since the framework enables for a systematic assessment of the wave propagation in both steady and unsteady conditions, namely to understand how we approach the steady-state solution (if it exists).

The reviewer also raises a point concerning spherical harmonics. In our analysis, we opt for Chebyshev polynomials (compared to spherical harmonics that use Legendre polynomials) as mentioned in the manuscript appendix. The choice of Chebyshev polynomials was motivated by their mathematical properties. Legendre polynomials are used in spherical harmonics but they necessitate numerical integration to assess the spectral coefficients, while in Chebyshev polynomials the spectral coefficients can be computed analytically or even by means of the FFT algorithm. This choice influences only the meridional derivative operators. In the zonal direction we also use the Fourier transform, so spherical harmonics and the proposed approach are equivalent in the zonal direction. We now clarify this point in the Appendix.

*L15: Hoskins and Ambrizzi (1993) should be stated, as it is probably the most classical reference in this context.*

Good suggestion. It is a very clear paper that we have now included in the revised version of the manuscript.

*L16: Wave guiding also goes back to the early work on refraction of Rossby waves, so this sentence reads a bit redundant in the light of the previous sentence.*

We agree with the referee that redundancies should be avoided but at the same time we think that the referred lines are necessary for readers that may not be acquainted with the waveguidability concept.

*L10-29: The first paragraph is rather long and the main topic is not clear. The paragraph might benefit from splitting it and more clearly addressing the context for this manuscript.*

We agree and we will rework the paragraph accordingly. The idea here was to motivate why Rossby wave propagation is important to understand extreme events as the latter are often occurring under special circumstances associated with amplified Rossby wave propagation.

*L31: Waveguidability is explained here for the first time, while the reader is left wondering during the first paragraph about its meaning.*

We now specify in the first paragraph that waveguidability relates to the "capability of jet streams to promote Rossby wave propagation". We then provide further details in the subsequent paragraphs.

*L61: Do the authors really mean "stability" in the sense of a wave instability or in the sense of applicability of the linear analytic solutions?*

We thank the reviewer for their remark. We are indeed referring here to stability in the sense of temporal wave (in)stability. The fact that we have an analytical solution is not related to wave instability. As briefly mentioned in the manuscript, we developed an additional linear code based on spherical harmonics and we solved it similarly to what we did with the nonlinear code. With both spherical harmonics solvers the analytical solution was not at hand and only a numerical assessment was done. For certain conditions (for instance, when the single-jet speed exceeded 20 m/s), the solution of the linear system diverged to infinity, exactly as and when predicted by the proposed model. However, by means of a modal analysis we are aware of all the possible unstable modes (if more than one are present) and therefore we have more understanding about the flow evolution. However, this stability feature and the corresponding divergence, can be even investigated numerically (for instance by simulating sufficiently long in time until the instability grows), although here we preferred an analytical tool since it was at hand.

*L74: It is confusing to refer to Lambda as longitude, which is not even used in the equations thus far, while at the same time using $\lambda_r$ as a dampening parameter. The authors are encouraged to change the naming of the dampening parameter to avoid confusion.*

We agree, we have now changed $\lambda_r$ to $\chi$ in the revised version of the manuscript .

*L147-153: Almost everything stated here is not new, even though the authors make it sound like a new discovery. Previous findings should be clearly stated and referenced.*

As the reviewer points out, the facts stated in these sentences are not new. We strongly disagree about the comment that we are making them "sound like a new discovery", and indeed in the original text we stated that these are "well-known analytical results" and further underscore this by stating that "it is already known . . .". What we do is to use this well-assessed material to validate our model, namely to quantify the discrepancy between the analytic solution (known in the case of solid-body velocity, without jets) and the proposed model estimation of the dispersion relationship. For instance, the error in the dispersion relationship (namely the value of the computed $\omega$ for a given $m$) was within machine precision (namely $O(10^{-11})$), while the waves were coincident with spherical harmonics with similar accuracy. Everything included in the validation section follows the same spirit. To avoid misunderstandings, we removed the word "Interestingly" from the sentence at line 147.

*L163-173: It is not made clear to the reader why this resolution sensitivity study is performed and its relevance to the assessment of waveguidability.*

A numerical algorithm is convergent when it approaches a constant value for sufficiently high resolution. The quantification of "sufficiently-high" is not determinable a priori, and requires numerical experiments. Thanks to this convergence study, we could state that a resolution of at least 128 latitudes is required to achieve good results. Smaller waves should require even higher resolution.

---

## Referee Report (RR1)

Review of egusphere-2023-316-V2

"A linear assessment of barotropic Rossby wave propagation in different background flow configurations"
by
Antonio Segalini, Jacopo Riboldi, Volkmar Wirth, and Gabriele Messori

**Recommendation: Reject**

**General Comments:**

The reviewers made a significant attempt in responding to my concerns. However, considering the length of the response, which is almost as long as a short manuscript, one wonders why so much clarification was needed for a piece of work that was apparently deemed ready for peer-review. Furthermore, the revisions almost amount to this manuscript becoming a new manuscript and thus a new submission. For future submissions, the authors are encouraged to assess the significance, context, and clarity of the work more carefully before entering the peer-review process.

Regarding my general comment about the introduction not leading to the actual research question addressed in this manuscript and that it left one wondering what this manuscript is about, the authors responded: "we believe that an introduction should provide context and motivation for the work, beyond a simple list of points that will be addressed in the analysis." It is exactly that what the authors have not provided in their first version, i.e., the context and motivation of their work. The revised introduction is an improvement, but one still wonders about the relevance of, for example, resonance, for which the authors use an entire paragraph. Does their method address this challenge? If so, it should be pointed out in the introduction, otherwise it leaves the reader wondering about the relevance of this discussion on resonance. The authors also discuss extremes and the context to climate change in the introduction, which is not followed up in the rest of the manuscript.

My specific comment on L16 was not understood correctly. My point was that there has been extensive previous work on the concept of wave guiding, not only as recent as the last ten years. It was this context that I was missing.

The new abstract clearly states that the main thrust of the paper is a novel algorithm, which would imply that my original interpretation that this is piece of work is mainly a technical paper was correct. As indicated in my previous review, for such a more technical manuscript, it would be recommendable to resort to more technical journals, such as GMD.

While I find the method and results interesting, the still somewhat confusing presentation of arguments and rather technical character make it not suitable for WCD in my point of view.

---

## Author Response (AR2)

A linear assessment of barotropic Rossby wave propagation in different
background flow configurations
by
A. Segalini, J. Riboldi, V. Wirth & G. Messori

Comments to Reviewer #1:
*(the text of the reviewer is in italic)*

We are extremely grateful for the detailed review, which we genuinely
believe has helped to improve the explanations of what are a number of
technically or theoretically complex passages of our study. In the following
we address the reviewer's suggestions for improvement, and point out the
changes compared to the original manuscript. Parts that have been rewritten
or added due to comments by the referees have been highlighted in red in
the revised version of the manuscript.

> *In the current version, the motivation is clearer, and the con-
> nection between the new numerical method and the concept of
> waveguidability is clearer. I think the manuscript could fit for
> publication in WCD, after another round of revision.*

We thank the reviewer for the support to our work.

> *1) Consistency of terms. (a). There are 3 types of models used
> in this study. Their names, according to the legend in figure 2a
> are: Linear Chebyshev, Linear SHT and Nonlinear SHT, where
> "SHT" stands for spherical harmonics transform. However, these
> names are not used consistently throughout the paper. The linear
> Chebyshev method is often called "linear" (e.g., line 186, caption
> of figures 1 and 3), which may confuse it with the linear SHT
> method. The nonlinear SHT method is simply called "nonlinear".
> The authors should choose a name for each method and use it
> consistently.*

This is a good suggestion. To avoid weighing the text with repeated
references to SHT and Chebyshev, we now clarify explicitly at the end of
Sect. 3 that we only discuss the linear Chebyshev and nonlinear SHT in
the main text, and that these are referred to as "linear" and "nonlinear"
simulations throughout.

*1) Consistency of terms. (b). Waveguidability is sometimes called "normalized meridional enstrophy density", though it is not made clear if these are two terms for the same physical variable. Specifically, is the expression in equation (17) the same as the waveguidability presented in figure 2?*

We thank the reviewer for pointing out this important point. We define the normalized meridional enstrophy density $E$ (Equation (19) in the current manuscript version) so that it can be used to obtain a waveguidability metric (Equation (20)), which is apt to compare the waveguidability of jets located at different latitudes. In these steps, it is important to separate the waveguidability metric (the tool) from the waveguidability (the concept), in the sense of the capability of the large-scale flow to zonally duct Rossby waves. Thus, the long name given to $E$ is a precise way to describe the latter quantity, but it is not the new waveguidability metric we propose.

The assessment of waveguidability shown in Fig. 2b employs the metric proposed by Wirth (2020), and stands as a proof of concept of the capability of the Linear Chebyshev method to reproduce previous results. The definition by Wirth (2020), however, becomes problematic when the waveguidability has to be assessed for jet streams located near the pole, since the smaller and smaller physical distance between the source and the receiver artificially inflates the value of the metric. We deem the metric in Equation (20) as a better assessment of the waveguidability property, since it is a number between 0 and 1, ranging between the two extrema of no waveguide and of a "perfect" waveguide.

*1) Consistency of terms. (c). The variable $\varphi_0$ is sometimes used to denote the latitude of the forcing (line 209) and sometimes – the latitude of the jet (caption of figure 3). The latitude of the forcing is also called "$\varphi_F$" (equation 15) and the latitude of the jet is also called "$\varphi_J$" (equation 16).*

As correctly noticed, there are three latitudes involved in our analysis:

1. the forcing latitude, indicated as $\varphi_F$ in equation (16);

2. the jet latitude, indicated as $\varphi_J$ in equation (17);

3. the latitude where we compute the normalised meridional enstrophy anomaly, indicated as $\varphi_0$ in equation (19).

In principle one can compute the enstrophy anomaly at every latitude with or without a jet ($\varphi_J \neq \varphi_0$ or $U_J = 0$) and with or without forcing at the same latitude, $\varphi_F \neq \varphi_0$. However, the idea with equation (19) was to introduce a perturbation at the same latitude $\varphi_F = \varphi_0$ to create an enstrophy source and use equation (19) to assess how much enstrophy remained at that latitude (within a small latitude band) and how much escaped. Since we focus on jet streams and expect that these might act as waveguides, we also chose the jet latitude as $\varphi_J = \varphi_0$. Therefore, jet latitude, forcing latitude and latitude where we compute the enstrophy anomaly coincide almost all the time, since we are trying to assess how strong a jet stream is by evaluating its capacity to hold the enstrophy (injected at the same latitude of the jet stream) within itself. An exception is for the double jet configuration, where there are two $\varphi_J$ that are different from $\varphi_F$. We have added a sentence to clarify this delicate point in the revised version of the manuscript in section 5.1 and explicitly state that for a single jet the three latitudes ($\varphi_0$, $\varphi_J$, $\varphi_F$) coincide.

> *1) Consistency of terms. (d). The term "temporal coefficient" (e.g, caption of figure 5, line 412) is used interchangeably with the term "principal component" (line 263) or simply "amplitude" (caption of figure 6). Please explain if these are all words to describe the same thing. If so – please be consistent with the terminology. If not – please explain what "temporal coefficient" means.*

Thank you for noticing this inconsistency, the two terms indeed refer to the same thing and we have uniformed our terminology to only use "temporal coefficients" in the revised version of the manuscript.

> *1) Consistency of terms. (e). The bar (over-line) is used sometimes to denote the background flow (equation 1) and sometimes to demote the amplitude of a wave mode (line 124).*

That's correct, thank you for noticing it. We have now resolved this issue in the revised version of the manuscript. The overline indicates now the background flow vorticity and background streamfunction only, while the amplitude of the wave mode is indicated by a tilde.

*2) Missing details. (a). For most of the results presented (at least figures 3, 4, 5 and 8), it seems that the authors used the same latitude for the jet and for the forcing, however this is not mentioned explicitly and the reader is left to guess what the latitude of the forcing is. Specifically, it is confusing the $\varphi_0$ is initially used for denoting the forcing latitude, while the jet latitude is sometimes called $\varphi_0$ and sometimes $\varphi_J$ (see comment 1c above). Also, it would help if the authors explain why they chose to use the same latitude for the jet and for the forcing (when this is the case).*

We have added a discussion about this point in the revised version of the manuscript in section 5.1.

*2) Missing details. (b). It is not described how the EOF analysis is done exactly. Specifically, I would expect to find an exact explanation for how the "temporal coefficients" shown in the bottom panels of figure 5 were calculated. If these are the same as principle component time series of the EOFs, then it is surprising that the principle component of the first EOF oscillates around 1 and not around 0. Usually, an EOF analysis is performed after removing the trend from the time series. Is that the case here or not? Please explain.*

No de-trending was applied to the present simulations. The time average of the meridional velocity field, $V$, was not removed and therefore the first EOF mode is approximately given by the time average of the solution. The EOF analysis was performed by considering the simulation results after 10 days from the start to exclude the initial transient (the wave generation and spreading) and the subsequent 90 days were collected and analysed by means of the singular-value decomposition (SVD) algorithm.

The inclusion of the time average in the SVD process affects only weakly the subsequent modes since the first vector of the decomposition is approximately provided by the arithmetic mean, while the second mode must be orthogonal to the first mode and so forth. This orthogonality constraint can influence the shape of the second mode (and subsequent ones), but the dimension of the space is sufficiently large to not have a noticeable consequence. On the other hand, the advantage of including the mean is that the field we reconstruct is complete and the time average is also projected into

orthogonal directions, facilitating the construction of a reduced-order model. We have now specified this methodology in section 4.2.

> *2) Missing details. (c). According to the caption of figure 8a, the green line marks the "locus beyond which the temporal vorticity variance of the nonlinear simulation becomes 10 times larger than in the stable regimes". While this is explained in the figure caption, this criterion is not mentioned explicitly in the text, when the "stability" of the nonlinear solution is considered (lines 291-294). I would expect a more detailed description and explanation for this criterion in the text. How is the temporal vorticity variance calculated? Which stable regime is it compared to?*

The green line is a result of several nonlinear simulations performed for many jet latitudes and jet strengths. We expected that all the points to the left of the solid black line in Fig. 8a (the linear neutral curve) should be characterized by a flow that just converges towards the equilibrium solution (i.e., all transient Rossby waves decay in time in this region), while all points to the right correspond to an exponential deviation from the equilibrium solution (i.e., Rossby waves amplify). This is what we see from the linear eigenvalue analysis with the linear Chebyshev method, but also by running linear time simulations with spherical harmonics. However, the nonlinear simulations do not behave like that: the jet must actually be stronger than the value indicated by the solid black line, in order to see the unsteady waves grow indefinitely.

The temporal variance of the meridional velocity was computed from the nonlinear time simulations to diagnose such a growth. If the simulation converges in time, the temporal variance remains low, while the latter increases significantly when an unsteady behavior is present that is not damped. This is indicated in figure 1 of the present response where the variance is reported for different jet conditions. It can be clearly seen that nonlinear simulations are more stable than linear ones. The threshold has been decided arbitrarily to $\langle (v - \langle v \rangle)^2 \rangle = 2 \text{ m}^2/\text{s}^2$ to mark a sufficiently strong growth. This has been clarified in the revised version of the manuscript at lines 238-239.

> *3) Interpretation of the nonlinear simulations. (b). The authors interpret the behavior of the nonlinear solution presented in figures 5 and 6 as evidence for a limit cycle (e.g. line 265, 294).*

[Figure]

Figure 1: Meridional velocity standard deviation for different jet latitudes and strengths from nonlinear simulations. A logarithmic scale is used in the color scale. The white dashed line and white solid line indicate the Rayleigh criterion and the linear neutral curve, respectively, while the red solid line is the locus where $\langle (v - \langle v \rangle)^2 \rangle = 2 \text{ m}^2/\text{s}^2$.

*I am skeptic about this interpretation. First, because in order to identify a limit cycle, a phase space should be defined and the existence of the limit cycle and its stability need to be shown in this phase space. I don't see what is the phase space in which the authors find a limit cycle. Second, I disagree that the wave modes shown in figure 5 are not traveling waves. In lines 263-265 the authors argue that "The trajectories... behave as traveling waves: however, the linear stability analysis does not support this interpretation as the waves should grow exponentially in magnitude because they are unstable". In this argument the authors ignore the fact that in the nonlinear simulation the mean flow is modified by the wave fluxes and therefore it can be stabilized. Additionally, the wave-wave interactions introduce a dissipative effect. So there is no reason to believe that these are not traveling waves (actually figure 6 shows exactly that these are traveling waves).*

We thank the reviewer for this comment, which gave us the opportunity to verify the presence of a limit cycle and to fortify the results we draw from our analysis.

By traveling wave we meant a neutral solution of the equations where its amplitude does not change in time. The linearised analysis did indicate the presence of unstable modes for jet speed 40 m/s, that however did not amplify exponentially indefinitely. The reviewer is right about the fact that a nonlinear traveling wave that was neutral with respect to the new equilibrium point could have, in principle, emerged. Only the nonlinear system can be used to sort out this matter.

In order to assess whether a limit cycle or a traveling wave takes place, it is important to define a state space – as the reviewer points out. We ran a long-time (200 days) simulation starting from the background state (this is referred as the reference simulation in the following analysis). From this simulation we calculate the EOF modes of the streamfunction, namely the variable used to solve the barotropic vorticity equation. As in Fig. 5 of the manuscript, the first mode $\Psi_0(\mathbf{x})$ is close to the time average of the forced system, while the successive modes $\Psi_1(\mathbf{x})$, $\Psi_2(\mathbf{x})$, ... are interpreted as Rossby waves. The projection of the instantaneous streamfunction field $\Psi(\mathbf{x}, t)$ into the $n^{th}$ EOF mode provides the mode amplitude as

$$a_n = \int \Psi(\mathbf{x}, t)\Psi_n(\mathbf{x})\mathrm{d}\mathbf{x}\,. \tag{1}$$

The instantaneous values of the coefficients can be collected into a vector

$$(a_0, \, a_1, \, a_2, \, a_3, \, \dots)$$

that provides our state space. The EOF analysis of the reference simulation provided typical amplitudes of the EOF temporal coefficients $\sqrt{a_{n,ref}^2}$ and the EOF spatial basis, that is from now on kept constant.

We ran several nonlinear simulations with different initial conditions given by

$$\Psi(\mathbf{x}, 0) = \sqrt{a_{0,ref}^2}\Psi_0 + \alpha\sqrt{a_{1,ref}^2}\Psi_1 + \beta\sqrt{a_{2,ref}^2}\Psi_2 + \dots, \qquad (2)$$

namely by initiating our simulation from the time average field plus a set of Rossby waves with arbitrary amplitudes. If the state space evolution from different initial conditions will lead to the same orbit, a limit cycle is present, while if the orbit depends on the initial condition a traveling wave is present. Figure 2 shows the evolution of the $a_1 - a_2$ and $a_3 - a_4$ coefficients for four different initial conditions, that nevertheless lead to a state evolution that spirals towards the same orbit, supporting our claim about the presence of a limit cycle.

It appears that, if the modes have a too high amplitude, they will be damped, while for too small amplitude they will amplify as predicted by linear theory. This is particularly clear for the $a_1 - a_2$ coefficients, while the $a_3 - a_4$ coefficients undergo a transient growth first and decay afterward.

We thank again the reviewer for pointing out that we did not provide sufficient evidence to support our claim about the limit cycle in the previous version of the manuscript. We have now added an appendix to discuss this matter.

*3) Interpretation of the nonlinear simulations. (a). As far as I could understand, the nonlinear simulations solve equation (3). This equation includes wave-mean flow interactions and wave-wave interactions. In contrast, the linear equation (equation 7) includes the effect of the mean flow on the waves, but does not include the effect of the waves on the mean flow, therefore it neglects both the wave-mean flow interactions and the wave-wave interactions. In the discussions in the paper, where the nonlinear simulations are compared with the linear method results, the authors assume that the differences between the solutions arise from*

[Figure]

Figure 2: State space sections. (Top left) $a_1 - a_2$ coefficients, (Top right) $a_3 - a_4$ coefficients. (Bottom left) $\left(a_1^2 + a_2^2\right)^{1/2}$ temporal evolution, (Bottom right) $\left(a_3^2 + a_4^2\right)^{1/2}$ temporal evolution. Black circles: state space evolution over 4000 hours with initial condition $\Psi(\mathbf{x}, t = 0) = 0$. Blue: initial condition $\Psi(\mathbf{x}, t = 0) = \Psi_0 + 3\Psi_1$. Red: $\Psi(\mathbf{x}, t = 0) = \Psi_0 + 0.5\Psi_2$. Magenta: $\Psi(\mathbf{x}, t = 0) = \Psi_0 + 3\Psi_3$. Cyan: $\Psi(\mathbf{x}, t = 0) = \Psi_0 + 0.5\Psi_3$. The starting point of the lines is highlighted by a green asterisk.

*the inclusion of wave-wave interactions in the nonlinear model, while ignoring the wave-mean flow interactions (e.g., lines 187-188, 295-296, 387-389). I think this is a very fundamental issue. When wave-mean flow and wave-wave interactions are included, the equilibration occurs due to the combination of stabilization of the mean flow profile by the wave fluxes, and the dissipation by wave-wave interactions. While the authors mention the latter, they ignore the former, which could be very important.*

We now also briefly discuss the potential role of wave-mean flow interactions in explaining the differences between the simulations in Sect. 4.1 and 4.2, and acknowledge that we cannot separate the role of wave-wave and wave-mean flow interactions when comparing the simulations.

*3) Interpretation of the nonlinear simulations. (c). The comparison between the stationary solution of the linear equation and the time-mean solution of the nonlinear equation assumes that we should expect them to be similar (e.g. lines 250-253, 371-373). I think these two solutions capture a fundamentally different phenomenon. The linear stationary solution captures a stationary (i.e. zero-phase speed) wave, forced by the topography. The nonlinear time-mean solution captures a statistically steady state. The paragraph in lines 253-268 (as well as parts of the conclusions section) tries to explain the similarity between the stationary solution of the linear equation and the time-mean solution of the nonlinear equation in an unstable case. They argue that one should expect the nonlinear solution to diverge in the unstable conditions (e.g. lines 236-237). Their interpretation for the lack of instability in the nonlinear case is that the system reaches a limit cycle. I would argue instead that the system reaches a nonlinear statistically steady state, where the mean flow is stabilized by the waves (in the time-mean sense), and the waves are equilibrated in the sense that their life cycles give a net zero growth, when averaged over time. I would definitely expect to find traveling waves in such a solution. These traveling waves don't necessarily need to be identical to the most unstable modes of the mean flow. They could be neutral modes in the linear sense, but they need to be able to maintain the mean flow in a profile that*

*enables them to go through cycles of growth and decay (see for example DelSole 2004, Lachmy and Harnik 2016).*

The linear equilibrium solution is a result of a linearised analysis while the time average of the nonlinear solution is another entity, as the reviewer points out. In both the paragraphs indicated by the reviewer our statement is that "the averaged flow field in the nonlinear simulation resembles ... the unstable equilibrium state calculated from the linear method". We did not expect them to be the same, but the qualitative similarity is only a result of empirical evidence. The discrepancies can arise in the linearised approximation because

1. the linear model is linearised around the background state and not around the equilibrium state: this error is enhanced when the topographic forcing has large amplitude.

2. the linearised model is unstable: in this case the perturbation growth leads to some wave fluxes that transfer energy from the wave to the mean flow

The remedy to the first issue is to linearize around the time average instead, an approach done by one of the authors in (Matsubara M, Alfredsson PH, Segalini A. Linear modes in a planar turbulent jet. Journal of Fluid Mechanics. 2020 Apr;888:A26.). However, the linear analysis with a nonzonal background flow is more computationally expensive since the Fourier decomposition is not separating the individual waves anymore.

In principle we were expecting that, if the background flow is linearly stable, perturbations should decay and the flow should approach the equilibrium point again, with minor discrepancy due to the linearisation approximation, as in the case with no jet. However, we were surprised to see that this good resemblance kept being the case even beyond the onset of linear instability, probably because the wave-mean flow terms in the time-averaged vorticity equation $\nabla \cdot \langle \mathbf{u}' \zeta' \rangle$ remains bounded by our damping term (see our previous comment on the limit cycle).

Our aim while discussing the limit cycle in the conclusions was more oriented towards the perturbation evolution, since we were expecting the onset of turbulence motion rather than a quasi-periodic wave activity. The scenario depicted by the reviewer is correct. Our statement about the unstable modes from the linear analysis resembling the EOF modes was also speculative: however, it is true that the linearly unstable modes are the infinitesimal

waves expected to grow the most, although this does not automatically exclude the possibility on non-normal growth (transient growth) of other waves.

Once again the description of the waves dynamics around the statistical equilibrium point requires a more accurate analysis linearised around the statistical steady state with a model of the wave-mean flow terms, an approach that we have not yet attempted and that goes beyond the scope of the present work.

> *4) Interpretation of the stability analysis. Figure 8a shows the maximum of the imaginary part of the linear eigenvalues (i.e., the linear growth rate) normalized by the damping time scale. The dashed line marks the locus where the absolute vorticity gradient changes sign (the Rayleigh stability criterion). The authors argue that the distance between the line where the linear growth rate is zero and the dashed line in figure 8a shows that "the Rayleigh criterion provides a necessary but not sufficient condition for the onset of instability" (lines 282-285). In the conclusions section (lines 382-384) they argue that the Rayleigh criterion was not capable of detecting the onset of barotropic instability. I disagree with this interpretation, because the linear stability criterion could easily be adapted to incorporate the effect of the damping term, by examining the line where the growth rate is equal to minus the damping time scale (i.e. where the growth rate is equal to -1 in figure 8a). Note that this line corresponds to the dashed line, meaning that it is consistent with the Rayleigh criterion of instability. This is not a coincidence. When the wave equation includes a linear damping term, the growth rate is expected to be the same as the linear growth rate of a model without damping, minus the damping time scale. Therefore, the results are consistent with the theory of barotropic instability, where the Rayleigh criterion marks the state where the linear growth rate of a model without damping is zero, and when linear damping is added, the growth rate is reduced by the damping time scale.*

We agree with the reviewer on this point. However, it is still true that the change in sign of the potential vorticity gradient is not associated to an unstable regime, but rather to a less stable one. The analysis is reported here below, although it follows the arguments of the reviewer.

The analysis starts by considering the barotropic vorticity equation in perturbation form (Eq. 10 in the manuscript)

$$\frac{\partial \widehat{\zeta}}{\partial t} + \frac{imU}{\sin\theta}\widehat{\zeta} - \frac{im}{\sin\theta}\frac{\partial Q}{\partial\theta}\widehat{\psi} + \chi\widehat{\zeta} = 0 \quad \text{with} \quad Q = \overline{\zeta} + \frac{f}{\cos\theta}. \tag{3}$$

By introducing the modal ansatz $\widehat{\zeta} = \widetilde{\psi}e^{-i\omega t}$ it is possible to simplify (3) into

$$-(i\chi + \omega)\widetilde{\zeta} + \frac{mU}{\sin\theta}\widetilde{\zeta} - \frac{m}{\sin\theta}\frac{\partial Q}{\partial\theta}\widetilde{\psi} = 0. \tag{4}$$

By multiplying all terms with $\sin^2\theta\widetilde{\zeta}^*/(\partial Q/\partial\theta)$ and integrating in colatitude one obtains

$$-(i\chi + \omega)\int_0^\pi \frac{\sin^2\theta}{\partial Q/\partial\theta}\left|\widetilde{\zeta}\right|^2 \mathrm{d}\theta + m\int_0^\pi \frac{\sin\theta U}{\partial Q/\partial\theta}\left|\widetilde{\zeta}\right|^2 \mathrm{d}\theta - m\int_0^\pi \sin\theta\,\widetilde{\zeta}^*\widetilde{\psi}\mathrm{d}\theta = 0, \tag{5}$$

where the last integral can be rewritten since $\zeta = \nabla^2\psi$ leading to

$$\int_0^\pi \sin\theta\,\widetilde{\zeta}^*\widetilde{\psi}\mathrm{d}\theta = -\int_0^\pi \sin\theta\left|\frac{\partial\widetilde{\psi}}{\partial\theta}\right|^2 \mathrm{d}\theta - m^2\int_0^\pi \frac{1}{\sin\theta}\left|\widetilde{\psi}\right|^2 \mathrm{d}\theta. \tag{6}$$

The last two terms of Eq. (5) are indeed real and do not contribute to the instability (i.e., the imaginary part of $\omega$). The only imaginary term is

$$-(\chi + \omega_i)\int_0^\pi \frac{\sin^2\theta}{\partial Q/\partial\theta}\left|\widetilde{\zeta}\right|^2 \mathrm{d}\theta = 0, \tag{7}$$

Equation implies that either $\omega_i = -\chi$ or the integral must be zero, the latter happening only when $\partial Q/\partial\theta$ changes sign, namely the Rayleigh criterion. We have described this aspect in the paper in Sect. 4.1 without providing the equations to avoid shifting the focus on this aspect. However, as mentioned in the manuscript, the inclusion of the damping term does not inhibit the application of the Rayleigh criterion, but it limits that to the fact that when the absolute vorticity gradient changes sign the imaginary part of the eigenvalue can be different than $-\chi$, therefore still stable.

*5) Section 6 This section doesn't include a discussion of the implications of the results. I couldn't understand the motivation for looking at the time-dependent solution and what the conclusions from this analysis are.*

We have decided to remove this section since it was interesting to us only as an analytical solution of the linearized equations, without however providing additional insight into the atmospheric dynamics than what already obtained from the stability analysis.

*1) Line 84: Lambda is defined, but it is not used in equation (1).*

Since we are defining the colatitude $\theta$ and the latitude $\varphi$, it is natural to introduce the notation for the longitude there too, rather than later on at equation (4).

*2) Line 105: Since equation (9) includes variables with a "hat", denoting the amplitude of the Fourier components, it would be better to define the Fourier components (Psi-hat(theta, t)exp(imx)) here, before the equation, or at least mention what the hat symbol means.*

We have now fixed this point in the revised version of the manuscript by providing the Fourier transform definition (Eq.- (9)).

*3) Line 118: Something in the wording is not correct, "achieved at regime" doesn't sound right. What does it mean?*

We meant that equation (13) is the streamfunction field after sufficiently long time. Initially we denoted this as the infinite-time solution. However, when this equilibrium state is unstable, an exponential divergence from the equilibrium state is expected and equation (13) should never be observed: in the nonlinear simulations we do observe anyhow a similar field in the time averaged field instead. We have reworded this sentence to clarify the above interpretation.

*4) Line 124: The bar (over-line) was used before to denote the time-mean background solution, here it is the amplitude of the Fourier component in time.*

We have now fixed these inconsistencies in the revised version of the manuscript. The overbar denotes the background flow while the tilde is used to indicate the Fourier transform of the perturbation.

*5) Lines 137-138: This would be a good place to refer again to the appendix.*

Thank you for the good suggestion; we have now added a reference to the Appendix there.

*6) Line 141-142: The first sentence of this paragraph seems to belong to the previous paragraph.*

Thank you for spotting this error. We have now corrected and moved the sentence to the previous paragraph.

*7) Line 144: Please mention exactly which linear and nonlinear equations the SHT package solves. Are these equations 7 and 3?*

Yes. Three solvers were developed in this project. Two based on the SHT transform (one solving nonlinear equation (3) and the other solving the linear equation (7)), and one based on the Chebyshev formulation (solving only the linear equation (7)). Since the two linear solvers gave the same answer, the majority of the paper is based on the analysis of the linear Chebyshev code and the nonlinear SHT code. One could argue that it would have been more consistent to just do the entire work with the SHT method only: however, the Chebyshev methodology provides an analytical form for the derivative matrices (equation A4 in the appendix) so that the eigenvalue problem could be easily formulated. We have specified this more clearly in the revised version of the manuscript at lines 185-190.

*8) Line 153: Bar (over-line) was used before to denote the time-mean, but here it is used to denote the zonal wind divided by cosine latitude.*

Thank you again for the careful review. This issue has now been fixed by replacing $\overline{U}$ with $U_0$ or directly with 15 m/s.

*9) Line 167: created by -¿ forced by.*

Corrected.

*10) Line 183: Delete the second "for the".*

Corrected.

*11) Line 190: "between the forcing the the monitoring sector" –
the first "the" after "forcing" should be replaced by "and".*

Corrected.

*12) Caption of figure 2: It says that "N" is the number of lati-
tude/longitude grid points for the Chebyshev simulations and the
truncation number for the SHT method. Why not define N as the
number of latitude grid points for all the cases, to be consistent?*

We have modified the caption of figure 2 according to the suggestion of
the reviewer so that we use the number of latitudes in both simulation setups.

*13) Line 239: "...is also not be ruled out" – delete the "be".*

Corrected.

*14) The authors use expressions related to time when referring to
the behavior of the system as a function of the model parameters
(the word "after" in line 299 and the words "started" and "later"
in lines 387-388).*

We indeed used incorrect terms and we have now fixed this in the revised
version of the manuscript, replacing for instance "after" with "beyond".

*15) Line 300: "...the maximum growth rate in Fig.8b..." – fig-
ure 8b shows the wavenumber, not the growth rate.*

That is the wavenumber associated to the maximum growth rate. We
have now corrected this in the revised version of the manuscript.

*16) Line 310: "jets" – change to "jet".*

Corrected.

*17) Line 325: "...equally efficient waveguides" – why do you say
they are equally efficient if their waveguidabilities are not equal?*

What we meant was that the waveguidability is high in both cases, but the reviewer is right and we have reformulated the sentence to: "...similarly efficient waveguides".

> *18) Line 338: "...the waveguidability is reduced...". I assume the authors mean that in the double jet case it is reduced compared to the single jet case. However, this should be mentioned explicitly and not left for the reader to guess.*

Correct. We now state this explicitly in the revised manuscript.

> *19) Line 384-385: This sentence is not clear. Specifically, the phrasing of the part: "...the condition of instability corresponds to a first increase of the...".*

Here we wanted to highlight how the application of the Rayleigh criterion leads to a necessary but not sufficient condition for the instability. We have rewritten the concluding section to facilitate the reading of the manuscript so we hope that the overall clarity has improved.

> *20) Line 388: Please explain what were the signs of barotropic instability in the nonlinear simulations? Why do you consider them to be signs of barotropic instability?*

For the adopted extremely simplified simulation setup only a barotropic instability can take place. We noted and discussed in the stability section that the nonlinear simulations remain temporally stable even beyond the neutral curve (namely for a jet strong enough to trigger a linear instability), although beyond the green curve in Figure 4a, even the nonlinear simulations show an unsteady behavior that is not decaying in time and the overall flow field remained time dependent.

> *21) Line 401: Delete the "that" after c).*

Corrected.

> *22) Line 410: "these evidences" – change to "this evidence".*

Corrected.

*23) Lines 412-413: The sentence "the temporal coefficients could be determined by solving a small set of nonlinear ordinary differential equations" is not clear. What do you mean by "temporal coefficients"? Are these the same as the principle component time series (see major comment 2b)? Which small set of equations are you referring to?*

The EOF temporal coefficients are the main subject of this sentence. As specified in our previous reply, we have now uniformed our terminology throughout the text to always refer to these as "temporal coefficients". If we substitute the EOF decomposition of the form

$$\Psi(\mathbf{x}, t) = \sum_{n=0}^{N} a_n(t) \Psi_n(\mathbf{x}), \tag{8}$$

in the barotropic vorticity equation (3), and exploit the orthogonality of the modes $\Psi_n(\mathbf{x})$ it is possible to obtain a nonlinear system of ODEs where the unknowns are the coefficients $a_0, a_1, a_2, \dots$. This model is extremely cheaper than the original equation (3) since only a small set of ODEs are solved. The book of Holmes et al. provides a detailed description of this methodology as a reduced-order model. We have added a clarification of this methodology in the manuscript.

A linear assessment of barotropic Rossby wave propagation in different
background flow configurations
by
A. Segalini, J. Riboldi, V. Wirth & G. Messori

Comments to Reviewer #2:
*(the text of the reviewer is in italic)*

We appreciate the feedback regarding our manuscript. In the following
we address the reviewer's suggestions for improvement, and point out the
changes compared to the original manuscript. Parts that have been rewritten
or added due to comments by the referees have been highlighted in red in
the revised version of the manuscript.

> *The reviewers made a significant attempt in responding to my
> concerns. However, considering the length of the response, which
> is almost as long as a short manuscript, one wonders why so much
> clarification was needed for a piece of work that was apparently
> deemed ready for peer-review. Furthermore, the revisions almost
> amount to this manuscript becoming a new manuscript and thus a
> new submission. For future submissions, the authors are encour-
> aged to assess the significance, context, and clarity of the work
> more carefully before entering the peer-review process.*

The purpose of peer-review is not merely that of a one-way communi-
cation from the reviewer to the authors, but also for the authors, reviewer,
editor (and in the case of EGU journals the broader scientific community)
to engage in a discussion on specific scientific aspects of the submission be-
ing reviewed. Indeed, if the sole purpose was one-way communication, the
whole concept behind EGU journals publishing both reviewer comments and
author replies would be somewhat moot.

Some topics, notably those of a more theoretical nature, may lead to
longer written discussions than others, due to the need to clearly explain tech-
nical aspects, assumptions and details that may be relevant to the framing of
the broader discussion. Moreover, the length of the replies is determined as
much by the authors as by the reviewers. Thorough reviews, raising relevant
points of discussion, will naturally lead to thorough replies, discussing those
points. A superficial review will likely elicit short and simple replies.

Coming to our specific case, we have received two very thorough reviews on our original submission, which we are grateful for. The fact that we disagreed with some of the reviewer's points, further contributed to a lengthy reply. Indeed, to facilitate the editor in their decisional role, we opted to provide particularly detailed replies to those points where we partly or wholly challenged the criticisms of the reviewer. Based on the above, we believe that judging the quality of a submission by the length of the replies to reviewers is a very poor call.

We further stand by our initial judgment that our manuscript is scientifically significant and clearly describes a set of new results. We are grateful for the time you have dedicated to reviewing our paper, as we are well aware it is an entirely voluntary and unremunerated undertaking, but disagree with your stance.

> *Regarding my general comment about the introduction not leading to the actual research question addressed in this manuscript and that it left one wondering what this manuscript is about, the authors responded: "we believe that an introduction should provide context and motivation for the work, beyond a simple list of points that will be addressed in the analysis." It is exactly that what the authors have not provided in their first version, i.e., the context and motivation of their work. The revised introduction is an improvement, but one still wonders about the relevance of, for example, resonance, for which the authors use an entire paragraph. Does their method address this challenge? If so, it should be pointed out in the introduction, otherwise it leaves the reader wondering about the relevance of this discussion on resonance. The authors also discuss extremes and the context to climate change in the introduction, which is not followed up in the rest of the manuscript.*

We wish to keep the discussion of the first version of our manuscript as short as possible, since this is not what is being reviewed here. We nonetheless wish to point out that our original introduction was structured as follows:

1. General background on atmospheric wave propagation (broad topic);

2. Relevance for surface weather (practical implications of studying the topic);

3. More detailed background on waveguidability and existing knowledge gaps;

4. Research question being addressed and structure of the paper.

We see no lack of contextualization in this structure.

Coming to the revised version, the new paragraph on wave amplification and resonance (we note that resonance takes up less than half of the paragraph) was restructured and expanded following the suggestion of this reviewer to split a paragraph in the original text. Splitting a paragraph seldom leads to a shorter text. In the introduction, we explicitly address the relevance of our analysis for wave resonance (lines 25-32). We agree that the single sentence on climate change is superfluous, and have removed it in the new version of the manuscript.

> *My specific comment on L16 was not understood correctly. My point was that there has been extensive previous work on the concept of wave guiding, not only as recent as the last ten years. It was this context that I was missing.*

We cite several studies from the 70s, 80s and 90s, so we still struggle to correctly understand the point the reviewer is making. It would be helpful to a constructive communication if the reviewer were to point to specific bodies of work that they think are lacking from our introduction.

> *The new abstract clearly states that the main thrust of the paper is a novel algorithm, which would imply that my original interpretation that this is piece of work is mainly a technical paper was correct. As indicated in my previous review, for such a more technical manuscript, it would be recommendable to resort to more technical journals, such as GMD. While I find the method and results interesting, the still somewhat confusing presentation of arguments and rather technical character make it not suitable for WCD in my point of view.*

The authors of the study have noticed – albeit without any factual data to support it – a trend towards fewer and fewer technical/theoretical contributions in climate dynamics journals, in favour of papers performing statistical or climatological analyses of large climate datasets. This may have

involuntarily led to the notion that more technical analyses are ill-suited for dynamics journals – something that we disagree with.

Concerning the focus of the study, we note that notable previous contributions on algorithms to study wave propagation have been published in WCD (Wirth 2020), J. Geophys. Res.: Atmos (Manola 2013) and J. Atmos. Sci. (Hoskins and Ambrizzi, 1993). A good rule of thumb to judge the relevance of a study for a particular sub-field is to look at its bibliography. In our case, none of the studies we cite has been published in GMD nor in JAMES, currently the two leading Earth System modelling journals. The journal we cite most often is J. Atmos. Sci., whose scope very much overlaps that of WCD.

To further address the concerns of the Reviewer, we emphasized in the manuscript, we emphasized in the manuscript the physical insights that can be gained from the analysis, which leads to an improved understanding of the idealized simulations used to study waveguidability in previous literature (such as Wirth 2020). In particular, we re-arranged the presentation of the results by grouping them in two new sections, one about results concerning barotropic Rossby waves and the other about results concerning the waveguidability problem. The structure of the conclusions was also modified to reflect the novel exposition of the results. We believe the knowledge gained will prove useful in designing future research efforts to study Rossby waves and their waveguides.

---

## Author Response (AR3)

A linear assessment of barotropic Rossby wave propagation in different
background flow configurations
by
A. Segalini, J. Riboldi, V. Wirth & G. Messori

Comments to the Reviewer:
*(the text of the reviewer is in italic)*

We appreciate the new feedback regarding our manuscript. In the follow-
ing we address the reviewer's suggestions for improvement, and point out the
changes compared to the original manuscript. Parts that have been rewritten
or added due to comments by the referee have been highlighted in red in the
revised version of the manuscript.

> *Following my comments, the authors added an appendix (B) to
> show that a limit cycle exists in the nonlinear simulations. This
> helped me understand what limit cycle they referred to in the pre-
> vious version, but I see it as an over-complicated way of looking
> at a simple phenomenon. Traveling waves and the limit cycle are
> in fact the same thing, as I elaborate in major comment 4 below.
> In any case, I think appendix B is not necessary, and this discus-
> sion deviates from the main theme of the paper, at least as I see
> it. I also still think that the authors are missing a much simpler
> interpretation of the results of the nonlinear simulations, based
> on the role of wave-mean flow interactions. I elaborate on this in
> major comment 2 below.*

Since this limit cycle analysis has been quite controversial and it leads
to a description similar to traveling waves, we have decided to follow the
suggestion of the reviewer and we have removed appendix B and dropped
the discussion about limit cycles.

> *In the bottom line, even though the paper does present a method
> that could be useful for future studies, and demonstrates its use-
> fulness, I think that some parts of it are written in a way that will
> not be understandable to the potential readers, and are presented
> in an over-complicated way.*

We hope that the revised version of the manuscript is more understand-
able since we have eliminated some of the more technical discussions.

**Major comments**

*1) A general comment on section 4: It is hard to follow how this section is connected to the main theme of the paper. If I understand correctly, the purpose of this section is to compare the prediction for the steady state solution (and its stability properties) based on the linear model with the "real" solution of the nonlinear system. If this is indeed the case, that should be stated explicitly. I think that the analysis of the limit cycle in subsection 4.2 and appendix B is not necessary, because I don't see how it contributes to the goal of this study. Perhaps all of subsection 4.2 is unnecessary. In any case, its title "dynamics of the unstable modes" does not describe the content well, because the modes of the nonlinear simulation are not unstable. If the authors choose to keep subsection 4.2, I think they can shorten it and remove the limit cycle analysis, for the reasons elaborated in comment 4 below.*

A linear analysis is useful as long as all the modes decay since waves initiated by a perturbation from the background state (or the topographic forcing in our case) vanish in time and a steady solution can be obtained (in our case equation (11)). If at least one mode is unstable, the linear analysis becomes much less useful for the assessment of the steady state. Beyond the neutral curve equation (11) should be of no use since the field should be time dependent and the traveling waves could orbit around another equilibrium state. Nevertheless, the good agreement between equation (11) and the time average of the nonlinear simulation continues to be the case and this required a deeper investigation into the dynamics observed in the nonlinear simulations, the only reliable information beyond the neutral curve. Section 4 is planned by following this red thread. We have now highlighted this aspect in the revised version of the manuscript and changed the title in section 4.2 to "Waves evolution in the linearly unstable regime" as pointed out the the Reviewer.

*2) The authors mention the role of wave-mean flow interactions only briefly and refer to it as a dissipative effect (lines 211-212). In other places it is ignored (lines 242, 282, 388-389). I commented on the previous version that a nonlinear simulation is*

*expected to reach a statistically steady state with a mean flow that is stabilized (or neutralized) by the effect of the waves on the mean flow. It's not a dissipative effect on the waves, it's a stabilizing effect on the mean flow. In their response, the authors mentioned that "the linear model is linearized around the background state and not around the equilibrium state". They say that "the remedy is to linearize around the time average instead" but claim that "the linear analysis with a nonzonal background flow is more computationally expensive". I agree, but I don't see why not take the time average of the zonal mean flow in the nonlinear simulation and perform the linear analysis on that. I would expect to see that it is more stable than the imposed background flow. If that is the case, it would show that the nonlinear solution does not diverge because the mean flow is stabilized by wave-mean flow interaction.*

We have reworded the sentence at line 212 mentioning the stabilizing effect of nonlinear terms instead. Whenever the linear simulation is analysed, these effects are ignored since they are not accounted for and they start to be taken into consideration whenever we observe significant deviations between linear and nonlinear simulations.

Regarding the linear analysis, it is not a problem to take the mean flow from the nonlinear simulation and perform a linear simulation around that to see what happens. However, it is harder to perform a linear stability analysis where the modes are obtained by means of an eigenvalue problem. This happens because we cannot anymore decompose the modes as independent waves with a fixed wavenumber in the zonal direction since zonal convolutions happen in the equations. We are currently working on a 2D eigenvalue problem to assess the Rossby waves on non-zonal background flows and we plan for a follow-up publication on the topic.

*3) Lines 226-231: Even though the authors agreed with my comment on the relevance of the Rayleigh criterion for the interpretation of the stability analysis, they have not changed anything in this paragraph. It stayed the same as in the previous version. This paragraph is difficult to read, as it is phrased in a very confusing way. Actually, a much simpler interpretation can be given, as I explained in my previous review. The Rayleigh criterion predicts*

*the borders of the stability region in the absence of dissipation. In order to adjust for the inclusion of dissipation, all you need to do is to subtract the dissipation time scale from the growth rate. The results in Figure 4 are consistent with this idea. Therefore, I don't understand why the authors chose not to mention this simple explanation. This comment applies also to lines 385-386 in section 6.*

We have re-written the above mentioned lines in the revised version of the manuscript. It is true that the linear dissipation just shifts the neutral curve towards higher velocities. However, our statements remain correct: the change in PV gradient (the non-dissipative Rayleigh criterion) is not immediately associated to the onset of barotropic instability.

*4) Here is why I think the limit cycle interpretation is an over-complicated way of representing traveling waves: Equation B1 defines the coefficients of the EOF modes (that are called "temporal coefficients" in other places). The authors claim in their response that these coefficients represent the amplitudes of the Rossby waves in the nonlinear simulation. I disagree, and I will try to explain why. Looking, for example, at modes 2 and 3 in figure 5, I see that each one of them represents a different phase of the same mode with wavenumber 5. Let's assume that this is a traveling wave. What would you expect the time trajectory to look like in the phase space defined by a2 and a3? It would look like a circle (as it does in figures 6 and B1a). This is because the traveling wave is equal to: a2(t)sin(m(x-x0))+a3(t)cos(m(x-x0)), where a2(t) = sin(omega\*(t-t0)) and a3(t) = cos(omega\*(t-t0)). Here I described, for simplicity, the wave shown in figure 52 as sin(m(x-x0)) and the wave in figure 53 as cos(m(x-x0)). The authors wrote in the response that "If the orbit depends on the initial condition a traveling wave is present". But actually, what happens is that during the simulation (no matter with what initial conditions) the flow adjusts toward a statistically steady state that supports specific traveling waves. In any case, this discussion does not add much to the main argument of the paper, so I don't think it is worth going into so much detail about the flow evolution. The way I see it, the nonlinear statistically steady state represents a*

*mean flow that was neutralized by wave-mean flow interactions, with traveling waves that, on time average, do not grow or decay. This interpretation is consistent with the results presented in this paper and with previous studies (I mentioned two of them in my previous review, and you can add to that a few papers by Brian Farrell). Lines 275-282 should be revised according to this comment.*

We agree with the reviewer that the limit cycle complicates the paper unnecessarily and we have dropped the limit cycle analysis in the revised version of the paper. We mention now those waves as traveling waves since, at regime, there is no difference between the two concepts. However, the limit cycle analysis provides an explanation about why the nonlinear simulation is not diverging, while the linear simulation diverges (in both the stability analysis and linear simulation approach with spherical harmonics). Following the Reviewer's suggestion, the papers from Hou & Farrell (1986) and Lachmy & Harnik (2016) have been included in the revised manuscript as they support the role of wave-wave interactions in maintaining an equilibrium mean flow.

*5) A general comment on section 5: It is not clear why the authors chose to compare between the waveguidability in the linear and nonlinear case only for the double-jet case (subsection 5.3), whereas for the single jet case (subsection 5.2) only the linear analysis is considered. The authors should at least provide a motivation for these choices.*

We have not performed any assessment of the waveguidability on nonlinear simulations in section 5. The only waveguidability assessment based on nonlinear simulations was shown in figure 2 just to compare with Wirth (2020).

*6) A general comment on section 6: I think that organizing the paragraphs as a list ("Firstly... secondly... thirdly") does not aid the reader to understand the overall theme of the paper. It would help to explain instead how each part is connected to the main goal and how all the parts combine to one story. For example, in line 382, instead of writing "Secondly, we elucidate some features of Rossby waves...", it would be better to motivate it by*

*explaining that the linear analysis is compared with the more realistic results of the fully nonlinear simulations, in order to assess its ability to capture what happens in the nonlinear simulations. The paragraph that starts with "Thirdly" does not connect to the subject of the previous paragraph – the comparison between the linear analysis and the nonlinear solution.*

We have reworded the conclusion paragraphs of the revised version of the manuscript.

**Minor comments**

*1) Lines 218-224: It's hard to follow this paragraph, because the text sometimes refers to the nonlinear simulations and sometimes to the linear solution, and it is not clear which is which. The nonlinear solution is mentioned and then it points to figure 3a, which shows the linear solution.*

We apologize for the lack of clarity. We have now rephrased the paragraph to highlight the various parts of figure 3 and motivate the linear analysis.

*2) Line 225: Is this sentence referring to the nonlinear simulations or to the linear solution?*

To both actually. One can perform an analysis by looking at the temporal solution of the linear/nonlinear problem, or by looking at the eigenvalues. The former approach is typical of complex systems that do not allow for a modal analysis at a reasonable cost. We have now specified this in the main text of the manuscript.

*3) Line 234: "the imaginary part of the most unstable eigenvalue" – why not use the term "maximal growth rate" here and in other places in the paper?*

This is a good suggestion that enhances the readability of the paper. We have now replaced imaginary part with growth rate throughout most of the manuscript.

*4) Line 235: "...has the same sign" – add after this "throughout the domain", or change to "...has a uniform sign".*

We have now changed the sentence to "a uniform sign throughout the domain".

*5) Caption of figure 4: "(equivalent to the neutral curve in the nonlinear case)" – this is not the appropriate phrasing. You are using an arbitrary threshold for defining a "neutral curve", so "equivalent" is not the right word. You could replace it with "(the chosen threshold for defining the neutral curve in the nonlinear case)".*

We have now replaced the sentence to "the chosen threshold for defining the neutral curve in the nonlinear case".

*6) Line 236: What do you mean by "stability margin"? You mention the range 15-22 m s$^{-1}$, but it's not clear what feature in the figure the reader should look at to see this.*

According to Cambridge dictionary a margin is "the amount by which one thing is different from another". We were mentioning that, while the PV gradient changes sign for some jet velocities, the onset of linear instability happens for larger velocities. We have now clarified what we meant with the velocity range by rewording the sentence to "above 15-22 m s$^{-1}$ (depending on the latitude) the growth rate of at least one eigenvalue becomes positive".

*7) Lines 238-242: One gets the impression that the threshold of 2 m$^2$ s$^{-2}$ represents an abrupt transition of the nonlinear simulations from low velocity variance to very high velocity variance (it says "increases drastically"). Is this really the case? If it is indeed an abrupt transition at this specific value, it would be good to show that in the figure. If not, then the phrasing should be changed to clarify that the threshold was chosen arbitrarily.*

The figure 1 below shows the meridional velocity variance from several nonlinear simulations for different jet positions/strength. The red line, indicating the locus with variance 2 m$^2$ s$^{-2}$, is also shown. Two color scales are used, one logarithmic when one sees the rapid increase in the variance (and

[Figure]

Figure 1: Meridional velocity variance in the nonlinear simulations for different jet velocity and jet latitude plotted. (Upper panel) logarithm of the variance. (Lower panel) linear value of the variance. The red line indicates the locus where the variance is 2 m$^2$ s$^{-2}$.

the red line is centered around that), and a linear one where the growth is visible and the red line is indicating more the start of the unstable region. The threshold of 2 m$^2$ s$^{-2}$ was chosen to sort out small amplitude variances in the stable region and it is arbitrary. We have clarified this aspect in the revised version of the manuscript.

> 8) Line 251: "which is indeed proportional to cos(phi)" – but the equation says that L is constant. I suppose you mean to say that this gives a wavenumber that is proportional to cos(phi).

Correct. We have corrected the sentence in the revised version of the manuscript.

*9) Line 252: "at different latitudes of the zonal jet" – add "of a given width".*

Done.

*10) Line 253-255: It is not clear how the first sentence, that relates to the degree of instability, is related to the second sentence, that relates to the wavenumber. The word "indeed" seems not appropriate here.*

The sentence has been changed to "This is verified in Fig. 4b that shows ...".

*11) Line 269: A reduced-order model of what?*

We have changed the sentence to "a reduced order model of the velocity field" in the revised version of the manuscript.

*12) Line 301: I assume the time averaging refers to the enstrophy and not to the vorticity anomaly. Please change the wording accordingly.*

Since the estimation is based on the linear stability results, there is no averaging involved, as done for instance by Wirth (2020). The equilibrium vorticity field is first subtracted by the background vorticity, the result is then squared and processed as described in the manuscript. We have clarified this aspect in the revised version of the manuscript.

*13) Line 302: The comment in the parenthesis is not clear. Why not use the time average? The statistically steady state represents fluctuations around a time-mean state, so an instantaneous state is not equal to the time average.*

Correct. We have removed the sentence.

*14) Line 307: It would be clearer if the words "when assessing how strong the jet stream is" were deleted.*

Done.

*15) Lines 314-315: It would help the reader if the authors point to the relevant features to look at in figure 8.*

We thank the Reviewer for this suggestion. We have added some sentences to clarify the motivation of the analysis. We have constructed figure 8 to show the main steps of our analysis. At the end a gradual waveguidability trend is present but at least now bounded between 0 and 100% providing a better metric than what available from previous definitions.

*16) Line 331: "Increases with jet speed from 0 to around 90%" – The jet speed values should also be mentioned, otherwise this statement is meaningless.*

There is a gradual increase of $W$ for all the simulated range, so the figure provides quantitative support to the qualitative text.

*17) Figure 8: Equation 19 is not expressed in percent, but in a dimensionless number between 0 and 1, but the variables in the figure are in percent. Please clarify this in the caption.*

Done.

*18) Line 343: Delete "once again". Also, the location of the forcing should be mentioned.*

Done.

*19) Lines 350-351: It should be clear if this calculation is for the linear analysis or for the nonlinear simulations.*

Right. We have clarified this point in the revised manuscript. We only use the linear method for the assessment of $W$.

*20) Lines 360-364: It is not clear where the forcing is located in each of the cases mentioned.*

We have clarified that at the beginning of the section in the revised manuscript.

**Language/typos**

*1) Line 75: "...in sections 4 and 5.3." – delete the "3" after "5."*

Done.

*2) Line 211: "The difference in waveguidability metric" – add "the" before "waveguidability".*

Done.

*3) Line 220: "some eigenvalues have in fact positive imaginary part" – delete "in fact" and add "a" before "positive.*

Done.

*4) Line 238: delete the word "time" before "variance".*

The variance can be computed in space too, so we feel that it is important to clarify here.

*5) Line 245: Change "associated to" -¿ "associated with".*

Done.

*6) Line 295: the word "metric" is written twice.*

Thank you for noting that.

*7) Line 351: Change from "jets with same velocity" to "jets with the same velocity, but different latitudes".*

Done.

---

## Author Response (AR4)

A linear assessment of barotropic Rossby wave propagation in different
background flow configurations
by
A. Segalini, J. Riboldi, V. Wirth & G. Messori

Comments to the Reviewer:
*(the text of the reviewer is in italic)*

We appreciate the new feedback regarding our manuscript and the careful
scrutiny of the reviewer. In the following we address the reviewer's sugges-
tions for improvement, and point out the changes compared to the original
manuscript. Parts that have been rewritten or added due to comments by the
referee have been highlighted in red in the revised version of the manuscript.

*1) Lines 230-233: This sentence was added following my com-
ment, that it is not a coincidence that the Rayleigh criterion curve
in figure 4a coincides with the growth rate of -1 (normalized by
the damping rate, $\xi$). I think this point should be written in a
more explicit way. The current text says that "... whenever
the absolute vorticity gradient changes sign, the imaginary part
of omega can be different than $-\xi$...". But it is not mentioned
that the dashed line in figure 4a shows exactly that the absolute
vorticity gradient changes sign when the growth rate equals $-\xi$.
This means that the Rayleigh criterion is relevant for the growth
rate, since it gives exactly a zero growth rate if no damping is
added, and the correction for the growth rate in the presence of
damping is exactly the damping rate.*

Thanks for suggesting this clarification: we have now pointed this aspect
out explicitly and slightly modified the wording to make the paragraph even
more understandable. Now it reads:

"At low $U_J$, the growth rate of the most unstable eigenvalue corresponds
to the dissipation rate $\chi$, which is negative (meaning that perturbations
are dampened with the same rate). This assertion is valid as long as the
absolute vorticity gradient has the same sign throughout the domain: as the
sign starts to flip (dashed line in Fig. 4a), a transitional regime is reached
where the eigenvalue growth rate is still negative, but not as low as $\chi$. This
is due to the fact that, in presence of dissipation, the simple change in sign
of the vorticity gradient (i.e., the Rayleigh-Kuo criterion) is not sufficient

to ensure the onset of barotropic instability. As $U_J$ increases, the stability margin decreases: above 15-22 m s$^{-1}$ (depending on the latitude) the growth rate of at least one eigenvalue becomes positive, leading to the divergence of the linear solution (bold black line in Fig. 4a). The nonlinear simulations, on the other hand, approach divergence for larger jet velocities than for the linear method: this is marked by the green line in Fig. 4a, denoting the locus where the meridional velocity time variance equals 2 m$^2$ s$^{-2}$ as an arbitrary threshold. Above such a threshold, the velocity variance increases drastically and it is concentrated at the jet location irrespective of the forcing, indicating the presence of barotropic instability. The neutral curve from the linear stability analysis provides then only a conservative estimate of the onset of instability: this underlines the stabilizing role of nonlinear terms (such as wave-wave and wave-mean flow interactions) not considered by the linear method."

*2) Line 234: "this results pinpoints" -> "this result pinpoints" (or "these results pinpoint").*

Done.

*3) Line 246: "such as wave-wave interactions" – I suggest to add also wave-mean flow interactions, as explained in my previous review. The same comment applies to line 395, where I suggest to change to "...pointing to potential damping or stabilizing effects operated by nonlinear terms (e.g. wave-wave and wave-mean flow interactions)".*

We agree and it has been changed in the revised version of the manuscript.

*4) Line 258: Following my comment the authors replaced "indeed" with "this is verified in...", but the problem remains that the previous sentence is not connected to the next sentence, i.e. looking at the wavenumber of the most unstable mode does not verify the analysis of Gill (1982) that relates to the growth rate rather than the wavenumber.*

Correct. We have rewritten the sentence with minor modifications to highlight that broad jets become stable while narrow jets are unstable at all jet latitudes.

*5) Line 284-286: "The nonlinearity of the waves was further confirmed..." – I didn't understand what this sentence is saying. Perhaps consider removing it.*

We have removed the sentence in the revised version of the manuscript. Our purpose was to highlight to the reader that we tested even different initial conditions but the regime wave amplitude remained the same.

*6) Line 290: "similarly to what suggested by..." -¿ "similarly to what is suggested by".*

Done.

*7) Lines 350-351: Is waveguidability assessed here around latitude 45? That's what it sounds like here, but later in lines 366-369 the waveguidability is compared with its values calculated around the jet latitude. This sounds like a strange comparison, where in one case the jet latitude and the forcing latitude are different and waveguidability is calculated around the forcing latitude, while in the other case the forcing latitude and the jet latitude are the same. If I misunderstood it, then perhaps it should be explained more clearly in the text. Otherwise, please explain the choice to calculate the waveguidability around the forcing latitude (when it is different from the jet latitude) and to compare it with a case where the forcing latitude is at the jet latitude.*

For the single jet case, the waveguidability is always computed at the jet latitude (where we located also the forcing). In the case of two jets, when we assess the waveguidability of one jet, we put the forcing at the same latitude of the jet. We added a sentence at line 357 to highlight that the waveguidability is computed at the jet latitude.

---

## Author Response (AR5)

A linear assessment of barotropic Rossby wave propagation in different
background flow configurations
by
A. Segalini, J. Riboldi, V. Wirth & G. Messori

Comments to the Editor:
*(the text of the reviewer is in italic)*

We appreciate the new feedback regarding our manuscript. We have
added a couple of sentences in that paragraph to highlight that

1. a very broad jet is stable

2. The latitude of the jet has a destabilizing effect (as in the case with
   $\sigma = 7°$ (stable in the tropics) and $\sigma = 5°$ (unstable everywhere))

3. a narrow jet becomes unstable faster at small wavelengths (or larger
   wavenumbers)

4. The wavenumber of the most unstable mode is associated to a constant
   wavelength, connecting to the previous paragraph.

We hope that this clarifies better the analysis. Alternatively, it is possible
for us to add a figure about the growth rate of the most unstable mode, but
we feel that such an additional figure will make the paper only unnecessary
longer.